



# Massive permafrost rock slide under warming polythermal glacier (Bliggspitze, Austria)

Felix Pfluger[1], Samuel Weber[2, 3], Joseph Steinhauser[1], Christian Zangerl[4], Christine Fey[4], Johannes Fürst[5], and Michael Krautblatter[1]

[1]Chair of Landslide Research, Technical University of Munich, Munich, Germany
[2]WSL Institute for Snow and Avalanche Research, SLF, Davos Dorf, Switzerland
[3]Climate Change, Extremes and Natural Hazards in Alpine Regions Research Center CERC, Davos Dorf, Switzerland
[4]Institute of Applied Geology, BOKU University, Vienna, Austria
[5]Institut für Geographie, Friedrich-Alexander-Universität Erlangen-Nürnberg, Erlangen, Germany

**Correspondence:** Felix Pfluger (felix.pfluger@tum.de)

**Abstract.** Recent studies have brought upon numerous evidence for enhanced rock slope failure from degrading permafrost rock walls. These failures have been thought to be subaerial and triggered by thermal heat propagation from rising air temperatures into the exposed rock faces. However, we have neglected that, at the same time, the dividing line between cold and warm basal states of polythermal glaciers has shifted some hundreds of meters upwards. This means that previously frozen and

ice-filled fragmented rock walls under cold glaciers have suddenly and for the first time in thousands of years been exposed to (i) hydrostatic pressures, (ii) warming and degrading ice in fractures, and (iii) rock mechanical degradation in warming rocks. One of the best case studies is the 3.9 to 4.3 million m$^3$ rock slide at Bliggspitze on 29 June 2007, which detached from a north-exposed, glacier-covered rock slope at $3200\,m$ above sea level. In this paper, we hypothesize that the transition from cold- to warm-based glaciers, a scarcely observed but widespread phenomenon, caused the massive rock slide. To prove this, we (a)

have analyzed the glacier transition since 1971 using aerial photographs coincident to meteo data, (b) compared 2013-2016 Ground Surface Temperature measurements to infer permafrost-prone/cold glacier thermal conditions, (c) categorized springs mapped in summer 2001/2012 according to geomorphological features and mineralization, d) performed Electrical Resistivity Tomography subsequent to failure on the destabilized rock flank in 2009, (e) conducted rock testing in frozen and unfrozen conditions and (f) modeled the mechanical impact of hydrostatic pressure, degradation of permafrost and glacier retreat in a

universal distinct element code (UDEC). Aerial photos indicate the existence of a cold glacier from 1971-2003 above the failure volume. On the rock face above the failure volume, ground surface temperature measurements demonstrate permafrost favorable conditions and underpin the presence of former and present cold-based glacier compartments. Since 2003, the warming of the Nördlicher Bliggferner Glacier has been evident in the lower and upper parts. In 2007, subsequent to the warmest January-June period in a 228-year temperature record in the area of Bliggspitze, the glacier opened massive ice crevasses above the

later rock slide, causing frequent ice-fall. New springs developed in the former permafrost flank some strong enough to cause debris flows. The high mineralization measured at springs at a proximal distance to the failure volume indicates active layer thaw. The inversion of Electrical Resistivity Tomography revealed several decameter deep-reaching thaw in the collapsed rock mass 2 years after failure. The tensile strength of tested paragneiss rock samples decreased by $-40\%$ from frozen to unfrozen



states, which reflects the mechanical degradation of rock bridges under warming permafrost. In this paper, we demonstrate a
new type of rock slope failure mechanism triggered by the uplift of the cold/warm dividing line in polythermal alpine glaciers,
a widespread and currently underexplored phenomenon in alpine environments worldwide.

## 1 Introduction

During the paraglacial transition, geomorphological activity increases in alpine areas worldwide (Gruber and Haeberli, 2007;
Ballantyne et al., 2014). Both permafrost degradation and glacier retreat alter thermal and hydrological regimes of rock slopes
challenging their mechanical stability (McColl and Draebing, 2019; Grämiger et al., 2020). Although the detachment areas
of high-volume rock slope failures in the mountain cryosphere are often in remote locations, the cascading effects of the
propagating mass movement can be severe, causing casualties and critical damages to infrastructure (Walter et al., 2020;
Shugar et al., 2021). With the observed warming of permafrost (Biskaborn et al., 2019) and volume loss of glaciers in recent
decades (Hugonnet et al., 2021), hazard source areas are expanding to higher elevations (Ballantyne, 2002; Huggel et al., 2012),
potentially reactivating dormant rock slopes.

In contrast to the relatively high frequency of low-volume failure events situated in zones of potential permafrost in the
Swiss Alps, the rare events of high-volume rock slope failures $> 10.000 \, m^3$ appear to not follow the observed pattern of
seasonality and elevation for low-volume events, such as the increase of frequency in summer at elevations above 2000 m asl
or in winter/spring at elevations below 2000 m asl (Phillips et al., 2017). Thus, high-volume failures in permafrost rock slopes
are difficult to predict.

Dating historic very high-volume rock slope failures in the Alps, Prager et al. (2008) pointed out a time-cluster some
thousand years after the beginning of the Holocene. These events are hypothesized to be prepared by glacial cycles and triggered
with a time lag to the Last Glacial Maximum (LGM) accounting for the loss in permafrost (McColl, 2012; Krautblatter et al.,
2013).

Over long time scales, glaciation cycles predispose rock slopes to failure by (i) over-steepening of valley flanks as a result of
glacial erosion (Holm et al., 2004; Mitchell and Montgomery, 2006), (ii) glacier advance and retreat causing mechanical stress
relief or increase to the bedrock, resulting in crack initiation and propagation in the underlying bedrock (Leith et al., 2014;
Grämiger et al., 2017) or even in first time slope failure by reaching displacements of several decimeters to meters (Fischer
et al., 2010; Rechberger and Zangerl, 2022), (iii) glacial hydrology exerting hydro-mechanical stresses in fractures (Grämiger
et al., 2020) and (iv) the sudden change of thermal regime during paraglacial transitions exerting thermo-mechanical stresses
to the rock mass (Grämiger et al., 2018).

In contrast, permafrost impacts the strength of rock slopes and intact rock properties, contingent upon its thermal state and
distribution. The observation of ice-filled joints at failure scarps in permafrost rock (Gruber and Haeberli, 2007; Zangerl et al.,
2019; Krautblatter et al., 2024) and its mechanical relevance were pointed out by laboratory studies on shear tests (Davies
et al., 2001; Günzel, 2008; Krautblatter et al., 2013; Han et al., 2023). Warming permafrost is found to be a key factor in
preconditioning rock slides along defined shear planes with ice-infillings: While warming, the shear strength of ice-filled joints





decreases by 17.2% for each degree Celsius increase in temperature within the range of $-4$ to $-0.5°C$ at an applied normal load of $0.8\,\mathrm{MPa}$ Mamot et al. (2018). Not only the shear strength of ice-filled joints, relevant for the stability of fractured rock mass, decrease distinctly while warming, intact rock properties indicate similar behavior: As temperatures rise towards

the freezing point p-wave velocities (Draebing and Krautblatter, 2012) and electrical conductivity (Krautblatter et al., 2010) experience a distinct drop before reaching a plateau at temperatures above $0°C$. Tensile strength decreased for relatively soft rock types such as sandstone/limestone by $67\%/61\%$ and for hard rock types such as granite by $6\%$ when comparing mean results of saturated intact rock samples tested at $-2.5°C$ and $+1.65°C$. Uniaxial compressive strength decreased for sandstone/limestone by $35\%/42\%$ and for hard rock types such as granite by $0.5\%$ when comparing mean results of saturated

intact rock samples tested at $-2°C$ and $+1.5°C$ (Mellor, 1973).

Permafrost acts as an impermeable hydrogeologic layer by sealing the rock matrix and fractures. The advective transfer of heat conveyed by percolating water substantially accelerates permafrost degradation in comparison to conductive heat flow alone. Favored by complex alpine topography, this process leads to the formation of deep thaw corridors along fractures, intersecting the permafrost body and enabling the build-up of high, transient hydrostatic pressures (Hasler et al., 2011; Magnin

and Josnin, 2021). During periods of exess melt water or precipitation, the transient rise in hydrostatic pressure is likely to destabilize greater slope volumes than rock- and ice-mechanical degratation under conductive warming alone (Gruber and Haeberli, 2007; Draebing et al., 2014).

To date, the impact of glacier activity and permafrost degradation on rock slope mechanics has been the subject of separate studies. However, the interactions between both have not been considered. The Bliggspitze (AT) rock slide is situated

in alpine permafrost and appears to be strongly linked to the activity of the Northern Bliggferner Glacier covering the slope before the first time formation of the rock slide. The geology and kinematics of the Bliggspitze rock slide have been investigated in detail by Zangerl et al. (2019), yet the processes leading to the rock slide remain unexplored. This paper focuses on the coupled effects of the glacier's thermal regime on permafrost and hydrogeology, and their mechanical implications, with the aim of investigating the processes destabilizing the 3.9 to 4.3 million m³, deep-seated rock slide at Bliggspitze in June 2007.


In this paper, we address three principal questions:

1. Can we decipher a cold-to-warm glacier transition using geophysics, temperature, and hydrological data?

2. How does the interaction of polythermal glaciers and bedrock permafrost destabilize rock slopes under climate change?

3. How does the change in thermal state affect rock slope mechanics using tensile strength as a proxy?

The study design is structured into three sections, each of which corresponds to the order of the research questions. The third section builds on the results presented in the previous sections. First, we investigate the evolution of the Northern Bliggferner Glacier in decades before the first time formation of the Bliggspitze rock slide, the permafrost conditions based on data recorded in the years after, and field observations and meteo records in the weeks immediately before the event. Secondly, we conducted tensile strength tests under frozen/unfrozen conditions and varying foliation orientation of paragneiss rock to con-

strain assumptions for the subsequent modeling study. Third, we conceptualized a mechanical model framework for the case of



the Bliggspitze rock slide. We hypothesize that the cold/warm dividing line of the glacier base is shifting to higher elevations under climate warming, affecting permafrost and hydrogeologic conditions. We investigate (i) the structural predisposition of the rock slope and analyze the impact of cryospheric forcings such as (ii) glacier unloading, (iii) permafrost degradation, and (iv) transient hydrostatic water pressure on rock slope mechanics by using a distinct element model.

## 2  Bliggspitze field site

Bliggspitze summit (3453 m asl, 46°55'5''N, 10°47'10''E) is located on the Kaunergrat ridge, which runs north-south and separates the Pitztal and Kaunertal valleys in Tyrol, Austria (for exact location, see map in Figure 1). Glaciers encircle the summit. The Eiskastenferner glacier is situated to the east, while the Northern and Southern Bliggferner glaciers are located to the west. The deep-seated rock slide occurred on the north-facing slope of Bliggspitze, predominantly covered by the Northern Bliggferner Glacier (Fig. 2a). A volume in the range of $3.9$ to $4.3 \cdot 10^6 \ m^3$ of glacier ice and rock mass was mobilized by the rock slide on 29 June 2007. Concomitant to the primary sliding process, secondary processes such as rockfalls, combined rock-ice avalanches, and debris flows have been observed hours/days before and after. The first time formation of the Bliggspitze rock slide caused the disintegration into individual slabs and created a continuous basal shear zone as indicated by the shear off set of the head scarp. Post-failure activity was evident in the months and years after the initial rock slope deformation. As the movement of the fragmented rock mass above the basal shear zone decelerated, various types of rapid movements with small volumes, such as rockfalls, debris avalanches, debris flows, and moderate movements, such as rock and soil slides, originated from the fragmented rock mass in the years after (process terminology after Cruden and Varnes (1996)).

The Northern Bliggferner's glacier outlines have remained relatively stable between 1969 and 2006, as shown in Figure 1. The Northern Bliggferner Glacier was divided into two sections, an upper and a lower one, with a substantial rock outcrop in the middle since at least 1969 (Fig. 2a). By comparing the glacier from historic maps of 1946 and ALS data from 2006, it was found that the Northern Bliggferner Glacier has experienced a significant loss in thickness, particularly in the lower section and slightly above the rock outcrop, with a reduction of several decameters (Zangerl et al., 2019).

The Bliggspitze rock slope is situated at the southern limb of a tectonic synform, with foliation of the paragneissic rock running sub-parallel to the slope. The rock mass is characterized by four distinct joint sets with spacing in the range of 0.2 - 0.6 m and steeply dipping fault zones at the top and middle sections of the slope (Dejean de la Bâtie, 2016). The head scarp was created by shearing off a prominent fault zone at 3270 m asl, resulting in a discrete shear offset of 40 m (Fig. 2b & c). The rock mass affected by the rock slide extends to an elevation of about 2900 m asl, most likely following a curved basal shear zone/rupture surfaces (for the structural geological profile, see Figure 3a). The existing fault zones were found to be crucial in the predisposition of the Bliggspitze rock slide. The fault zones at the Bliggspitze rock slopes can be characterized as soil-like brittle tectonic fault zones, with a high proportion of sheet silicate minerals (Mica group $20-30\%$, Chlorite group $10-20\%$), yielding in residual friction angles within the range of $25.7 - 28.9°$, as determined in laboratory tests (Strauhal et al., 2017). A comprehensive geological and kinematic model of the Bliggspitze rock slide can be found in the publication of Zangerl et al. (2019).





**Figure 1.** Overview of the study site, illustrating the dimension of the affected rock slope, historic glacier outlines, the position and results of measurements including Ground Surface Temperatures and electrical conductivity of springs, and profile sections of Electrical Resistivity Tomography and through the rock slide. Left map: Würmtal Valley, including the peak of Bliggspitze at the lower right (3453 m asl). Right map: Detailed zoom to the head scarp located at 3270 m asl. Glacier outlines are given for the Northern Bliggferner Glacier. The Southern Bliggferner is situated to the west of the summit of Bliggspitze, while the Eiskastenferner Glacier is located to the east. (Base map left: Orthophoto (2015-08), right: Slope map at the post-failure stadium (2007); DEM of 2006 and 2007 used for change detection provided by Land Tirol - data.tirol.gv.at)

## 3 Methodological approach

### 3.1 Pre-failure analysis of the Bliggspitze rock slide

The pre-failure stage defines the period before the first time formation of the Bliggspitze rock slide coinciding with the creation of the basal shear zone at the head scarp. The at-failure stage refers to the activity related to the initial deformation in the




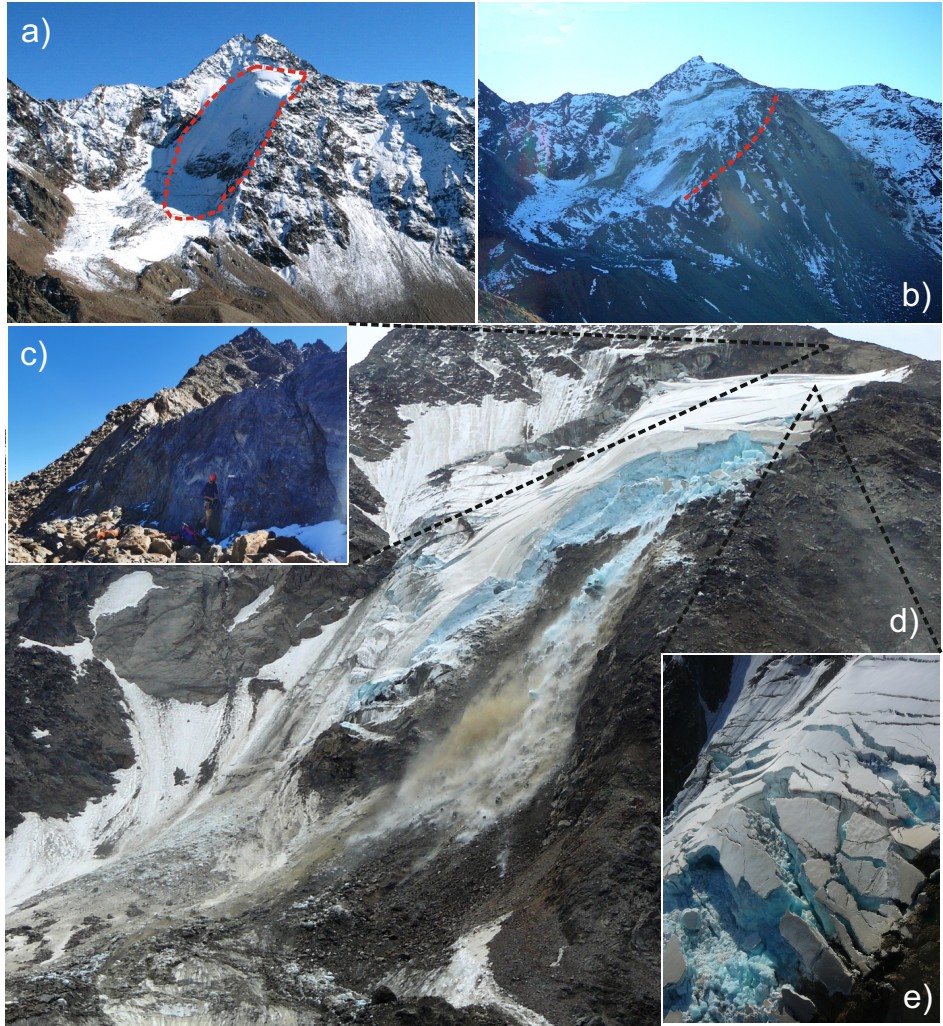

**Figure 2.** Photographs of the Bliggspitze rock slope taken of the pre-, at- and post-failure stadium showing a) an overview from the Northwest with the Northern Bliggferner glacier in the center - the glacier is divided into an upper and lower section by the rock outcrop in the middle (pre-failure stadium in Summer 2003), b) the post-failure stadium indicated by the fragmented glacier at the top and the sedimentation of debris at the lower part of the glacier (October 2007) - the inferred shear plane is illustrated with a dashed red line in the figure. Debris flows altered the west-facing slope in the years and days previous to the first time formation of the rock slide in June 2007, c) the basal shear zone at the head scarp with polished slickenside and the fragmented rock mass underneath, d) rock and ice falls (secondary processes related to the rock slide: at-failure stadium) and e) the widening of crevasses and fragmentation of glacier ice at the upper section favoring meltwater infiltration into bedrock (at-failure stadium). Dashed black lines indicated the location of more detailed, zoomed-in views. Photographs kindly provided by M. Krautblatter and G. Heißel.

days/hours before and after. Post failure stage is related to the on-going deformation of the displaced rock mass after initial failure along the basal shear zone (Leroueil et al., 1996).



Table 1 and 2 summarize time series and spatial data used to infer glacier evolution and permafrost conditions. Figure 1 shows an overview of the study site, including the extent of the rock slide and past glacial outlines, and compiles the position and results of investigations on permafrost at the site in a map.

**Table 1.** Meteorological and hydrological time series used to characterize pre- and post-failure conditions.

| time series | parameters | observation period | measurement interval | location | source |
|---|---|---|---|---|---|
| Histalp dataset 5x5 minute grid cell | T | 1780 - 2014 | monthly | Würmtal Valley (mid of cell) 2236 m asl | HISTALP dataset accesible via https://www.zamg.ac.at/histalp; Chimani et al. (2013) |
| Spartacus dataset 1x1 km grid cell | $T_{max}$,$T_{min}$, PPT | 1961 - 2022 | daily | Southern Bliggferner Glaciers (mid of cell) | SPARTACUS dataset accesible via GeoSphere Austria Hiebl and Frei (2016, 2018) |
| Hydrology of Vernagt basin | Q, T, elect. conductivity. | 2007 | 10 min. | 8 km distance to site 2640 m asl | Escher-Vetter et al. (2014) |
| Pitztal glacier meteo station | T | 2013 - 2016 | hourly | 8 km distance to site 2863.9 m asl | GeoSphere Austria, station id: 17315 |
| Brunnenkogel meteo station | T | 2013 - 2016 | hourly | 6 km distance to site 3437 m asl | GeoSphere Austria, station id: 173200 |
| Ground Surface Temperature measurements (GST) | T | hydrological years 2013 - 2016 | 2 h | at site, 3200 ± 20 m asl | Zangerl et al. (2019) |

Glacier changes were analyzed by mapping visual changes in historic orthophotos. In addition, we compared the rate of annual glacier elevation change for given periods during pre-, at- and post-failure stadium. For the pre- and post-failure stadium,

we calculated the rate of annual glacier elevation change via DEM-based raster calculation (1969 - 2000, 30 m resolution; 2007 - 2017, 1 m resolution, see Table 2). In regard to the at-failure stadium, we refer to the published dataset by Sommer et al. (2020) (2000 - 2012, 30 m resolution).

Permafrost conditions were inferred by Ground Surface Temperatures (GST) and Electrical Resistivity Tomography (ERT) in the area below the head scarp (for the location of the profile, see Fig. 1), the measured mineralization of springs and by field

observations suchs as ice-filled fratures.

For the recording of GST, 22 temperature loggers were placed in open fractures a few decimeters below the surface within the failed rock mass at an elevation of 3200 ±20 m asl. The loggers were covered with loose debris to avoid direct contact with snow cover and exposure to solar radiation. The applied HOBO U22-001 loggers recorded temperature every 2 hours from 2 September 2013 to 31 August 2016. We used daily standard deviation as a proxy to detect snow cover according to Haberkorn

et al. (2015); Draebing et al. (2022): Daily standard deviation > 0.5°C indicates snow-free conditions, whereas daily standard



**Table 2.** Spatial data used to characterize the cryosphere at the site.

| spatial data | year | file type/resolution | source |
|---|---|---|---|
| Glacier outlines | LIA, 1969, 1997, 2006 | shape-file | Austrian Glacier Inventory; Fischer et al. (2015) |
| Digital Elevation Models | 2006, 2007, 2017 | 1 m | Land Tirol - data.tirol.gv.at |
| | 2000 | 30 m | SRTM, Farr et al. (2007) |
| | 1969 | 30 m | DHM69, Lambrecht and Kuhn (2007) |
| Glacier Thickness Models | 1970, 2003 | 30 m | Sommer et al. (2023) |
| Rate of elevation change of glacier | 2000 - 2012 | 30 m | Sommer et al. (2020) |
| Orthophotos | 1969, 2003, 2007, 2009, 2010 | 0.2 - 0.25 m | Land Tirol - data.tirol.gv.at |
| Annual sum of incoming solar energy | 2013 / 2014 | 1 m | Land Tirol - maps.tirol.gv.at/ |
| Mapping of springs, measurement of electrical conductivity | 2011, 2012 | point-file | Dejean de la Bâtie (2016) |

deviation < 0.5°C indicates presence of snow cover that puffers air temperature / incoming solar radiation signals. Additionally, a daily standard deviation < 0.001°C was used as a threshold to detect the zero curtain. The data was initially published by Zangerl et al. (2019) and subsequently reanalyzed in the present study.

The ERT was conducted on 12 August 2009, two years after the first time formation of the Bliggspitze rock slide. For the
ERT survey, we used a Wenner array to measure 190 data points with a profile length of 100 m and electrode spacing of 2.5 m. Out of 190 data points, 155 were successfully coupled to the ground and were used for the inversion process with the software RES2DINV-4.10.3 (Loke and Barker, 1996). We conducted inversion under robust data constraint (cutoff factor: 0.005) for a refined model with a cell width of half the unit spacing. As the measured data contained considerable noise, we tested the inversion on a reduced data set after eliminating measured data points that do not correlate well with the modeled apparent
resistivity values. By fitting the model to 128 instead of 155 data points, we could essentially improve the root mean square (RMS) error to a value of $9.4\ \Omega m$ after the 7th iteration of the inversion.

We reprocessed data of 32 springs mapped in the area related to the Northern and Southern Bliggferner and Eiskastenferner Glaciers in summer 2011 and 2012 to potentially detect water originating from permafrost or glacier melt as a source (original data derived from Dejean de la Bâtie (2016)). Therefore, we classified springs according to the geomorphological forms in their
vicinity by the use of orthophotos and digital elevation models and categorized the measured values of electrical conductivity into two groups seperated by a treshold of $150 \cdot 10^{-6} Scm^{-1}$.





The chronology preceding the formation of the Bliggspitze rock slide was analyzed by utilizing climate, meteorological, and discharge data and examining the related events that occurred in the years and weeks before the rock slide (Table 1).

### 3.2 Laboratory experiments under frozen and unfrozen conditions

We conducted pseudo-Brazilian tests on intact rock samples under saturated, unfrozen, and frozen conditions and varying foliation orientation. Several decimeter-thick blocks of foliated Paragneiss (same lithologic unit as at Bliggspitze study site) were collected at rock outcrops next to the road along the "Kaunertaler-Gletscher Straße" at three different sites in elevation between 2000 and 2600 m asl.

In the lab, we prepared cylinders 50 mm in diameter and a length of 25 mm, maintaining a diameter-length ratio of $2:1$,
as suggested by Lepique (2008). Samples were prepared to be tested with angles of 0, 20, 45 & 90° between foliation and direction of the applied force. All samples were saturated at atmospheric pressure for 24 hours until reaching constant mass (DINDIN-EN-13755:2008-08 (2008)) and wrapped in a plastic layer to prevent dehydration. One-half of the samples were frozen until reaching a constant temperature of - 10°C and tested under the ambient temperature of $4 \pm 1°C$. The other half was kept and tested at room temperature $(20 \pm 1°C)$. To keep the ambient thermal influence on frozen samples while testing
as low as possible, samples were removed from the freezer immediately before testing and insulated with styrofoam. They were only exposed to ambient temperature during the tests. To keep track of the warming of the samples, we measured surface temperature and core temperature for specified samples. The core temperature measured at the center of gravity of insulated samples dropped from -10°C to -6.8°C at 5 minutes after removal from the freezer at no load applied.

Tensile strength was measured and calculated according to Lepique (2008) with modifications made to the measurement
device, resulting in the designation pseudo-Brazilian test (pBZT). Instead of using the recommended testing machine that allows for a continuous increase in load, we employed the manually operated Portable Point Load Test Apparatus developed by Wille Geotechnik, which is utilized for Point Load tests. The device with a force gauge to determine the breaking force was modified by substituting the cones with bridges to change the load type from point load to strip load. As the manual operating device was transportable, we set it up in a room with a constant ambient temperature of  4-6°C to keep external
thermal influences on frozen samples while testing low. The test duration was limited to max. 3 minutes. For testing, the cylindrical samples were mounted horizontally and fixed between the opposing bridges contacting the lateral surface. We rotated the samples along the cylindrical axis to test force application on different orientations of the foliation. Brazilian-tests were typically conducted using a constant strain rate. In our setup, we opted for a manually operated device that applies force through human power, which resambles a constant stress test. To ensure equal test conditions for all samples, tests
were conducted by the same person, with attention paid to increasing load continuously and consistently. A backtest (n=8) was conducted to compare the manually operated pBZT procedure with the standard Brazilian test procedure utilizing the recommended compression test machine according to Lepique (2008). The results demonstrated no statistically significant difference between the two methods.





### 3.3 Numerical discontinuum modeling of the Bliggspitze rock slide

We use the 2D mechanical modeling framework UDEC - Universal Distinct Element Code by Itasca Consulting Group (2019) - to analyze the mechanics triggering the first time formation of the Bliggspitze rock slide. UDEC uses the distinct element method to represent the rock mass by defined, discrete discontinuities, such as joints or faults and individual blocks resulting from the intersection of discontinuities. During simulation, those discontinuities (=contact areas of two blocks) act as boundaries that allow sliding, toppling, or rotating of individual blocks. The blocks can be represented by a continuum (zoning of

finite-difference element mesh) and are deformable according to assigned constitutive material models. We apply the Mohr-coulomb slip criterion in our modeling scenarios to represent shearing along discontinuities. We assigned linear-elastic (elastic, isotropic model) or elastoplastic (Mohr-Coulomb model) as constitutive material models to blocks representing rock mass or glacier ice respectively. The geologic 2D model of Bliggspitze (for location of profile see Figure 1, the structural geology is shown in Figure 3a) was used for the mechanical analysis.

#### 3.3.1    Implications of the glaciers' thermal regime change on permafrost and hydrogeology

The Bliggspitze rock slope was almost entirely covered by the Northern Bliggferner Glacier before the first time formation of Bliggspitze rock slide. The Northern Bliggferner Glacier, which ranges in elevation from 2800 to $3.200 \pm 20\ m\ asl$, excluded the ice apron above the Bergschrund that reaches elevations of up to $3360\ m\ asl$, and spans approximately one kilometer in length in 2006, can be characterized as a polythermal glacier of type D according to classification of Pettersson (2004): Cold

ice is restricted to upper-most accumulation area, whereas low altitudes exhibit warm ice. These glacier types are common in high altitudes of the European Alps (Suter, 2002). Permafrost and hydrogeology are strongly affected by the basal regime of the overlaying glacier. Inferred from observation of the evolution of the Northern Bilggferner Glacier, as shown in the section *results - Pre-failure analysis*, we drafted a generic model coupling the interdependencies of glaciers' basal regime, permafrost and hydrogeology. We define the border between a glacier's cold and warm base as the Polythermal Dividing Line (PDL, Fig.

3b). At elevations below the PDL, the glacier is warm-based, meaning basal discharge is possible and meltwater can infiltrate into the bedrock, where permafrost is absent. However, at elevations above the PDL, the glacier is cold-based, indicating the existence of permafrost and hindering surficial water from reaching bedrock. For simplicity, we assume that permafrost is bound to the elevation of the PDL. Especially in spring, when meltwater discharge reaches seasonal peaks, groundwater levels rise temporarily. We use the elevation of the PDL and slope topography as upper limits to model a free water table under

peak discharge conditions. Similar to a shift of Equilibrium Line Altitude, we hypothesize that the PDL is shifting to higher elevations under a warming climate too. We investigate the mechanic response of the rock slope to permafrost degradation and peak groundwater level, both simulated as a function of the elevation of the PDL, which was iteratively increased by $\Delta h_{PDL} = +30\ m$ after each mechanical simulation (Fig. 3b).





### 3.3.2 Scenario analysis

We developed four individual scenarios S1 to S4 (Fig. 3). In the first step, we evaluated the rock slope's predisposition to failure by testing the model with varying compositions of structural elements such as fault zones, joints, foliation, and basal shear zone (scenario S1). As a result of scenario S1, we progress with a model of representative geometry to assess the mechanical effect of glacier unloading (S2), permafrost degradation (S3), and peak groundwater level (S4) as potential triggers for failure. Figure 3c) summarizes the detailed parametrization of each scenario and the underlying assumptions. Subsequently, we describe each

scenario in detail.

**S1 - Structural predisposition**: In the first scenario, we assess which mapped structural features are relevant in controlling slope displacement. The given 2D-cross-section aligns with the direction of kinematic movement. The distribution and direction of fault zones and the basal shear zone were adopted according to Zangerl et al. (2019). For the foliation and the existing four joint sets, we calculated the mean apparent dip and projected it into the cross-section. Joint sets of which the strike forms a

sharp angle with the strike of the cross-section smaller than 60° were not considered, as these structures were not mechanically relevant. The model geometries A to F distinguish themselves by containing a different degree of structural complexity. We assumed that before the formation of the rock slide in 2007, the rock slopes' factor of safety must have been slightly $> 1$, representing a situation where the driving forces are only marginally smaller than the resisting forces. By varying pairs of friction angle and cohesion for each model geometry, we aimed to identify states of the rock slope that are close to failure.

For scenario S1, the assigned rock mass properties were calculated using values obtained by the Geological Strength Index - characterization (Hoek and Brown, 2019) and intact rock properties from laboratory experiments (Table S1). These values were kept constant for all further simulations.

**S2 - Glacier unloading**: We modeled the Northern Bliggferner Glacier as two elastoplastic blocks representing the upper and lower part (2a). Mass loss of the glaciers was simulated by assigning different densities to the modeled glaciers. The

material parameters of the elastoplastic blocks, as well as the shear parameter for the boundary between glacier and bedrock, were determined by sensitivity tests according to the following criteria: i) To account for the ductile behavior of ice, the glacier should exert minimal resistance against deformation when interacting with the rock mass. ii) While the glacier body does not deform under its weight, it should remain stable and not shear off the bedrock contact. Except for the density of glacier ice, the material parameters were kept constant throughout all stages (Fig. 3c). Based on the assumed original state the density of the

glaciers was doubled, halved, or the glaciers themselves were removed before the start of cycling to demonstrate the effect of glacial buttressing or loss of toe support on slope stability.

**S3 - Permafrost degradation**: We assume a uniform permafrost distribution above the Mountain Permafrost Altitude (MPA) where frozen rock prevails. Permafrost is absent below the MPA, and the rock is unfrozen. In the case of Bliggspitze, the MPA is equal to the elevation of the PDL (Fig. 3b, $h_{MPA} = h_{PDL}$). We incorporated the effect of permafrost by assigning

higher shear parameters to discontinuities within the frozen rock. In sub-scenario S3A, we run simulations with a surplus in friction angle of $\Delta\phi = +5°$ for frozen in respect to unfrozen rock, while cohesion c remained unchanged. In sub-scenario



S3B, we run simulations taking into account both, the effect of ice-filled joints $\Delta\phi = +5°$ and the effect of frozen rock bridges $\Delta c = +0.02\ MPa$ on shear strength of discontinuities within the frozen area.

**S4 - Peak groundwater level**: Scenario S4 investigates the impact of temporary peak groundwater levels initiated by an
excess supply of meltwater or rainfall. The elevation of PDL defines the horizontal boundary below which a free water table may exist within the rock mass (Fig. 3b, $h_{PDL} = h_{wl}$). The slope topography at elevations below the PDL defines the surface of the free water level, assuming fully saturated rock mass. In UDEC, hydrostatic pressure was exerted within all discontinuities below the water table throughout the full cycle time. According to the principle of effective stress, the total stress is diminished by the water pressure present at each specific location. Stress-related variation in joint aperture was ignored. Given
the dense fracture network of the rock slope, exhibiting loosened blocks and joint spacings up to decameters at surfaces, water permeability is likely to remain unaffected by variations in joint aperture. The mechanical response of the model did not influence the hydrogeological situation. Hydrostatic pressure was applied only within discontinuities; blocks were assumed to be impermeable.

### 3.3.3   Model initialisation and simulation workflow

Subsequently, we describe the initialization of the mechanical model. We refined the mechanical model according to topography and structural geological features, introducing them as discontinuities. The meshing of blocks was performed with a maximum edge length of 10 m. To enable the rotation of blocks and reduce overall execution time, the rounding length of corners was set to $0.5\ m$. The left, bottom, and suitable model boundaries were fixed with no-velocity conditions. The gravitational force was assigned to $9.81\ m/s^2$. To account for the topographic effect on stress distribution, we initialized models
with a vertical in-situ stress ratio of $k = 0.5$ ($0.5\ \sigma_y = \sigma_x = \sigma_z$). For initializing a base model serving as a starting point for the simulation of each single scenario, we applied shear strength values of $\phi = 40°$ and $c = 0.5 MPa$ to discontinuities and bulk/shear modulus of $K = 37\ GPa\ /\ G = 14\ GPa$ to the blocks being modeled as linear elastic entities. The base model was then solved until equilibrium was reached, after which deformations were reset to zero. The calculated stress state of the base model represents the initial condition for the scenario simulations.

The procedure for analyzing individual scenarios based on the initialized and properly parameterized model is described below and illustrated in Figure S1. For each simulation, we ran 10.000 mechanical timestep cycles and recorded 9 different histories at monitoring points within parts of the rock mass and the glacier that are either expected to be displaced or stable. We visually compared the displacement results of the overall model at the end of cycling and inspected the displacement as a function of the mechanical timestep cycled for the selected monitoring points. If the displacement of the unstable parts within
the rock mass approaches an equilibrium state by the end of cycling, meaning after initial settlement, the slope restabilized itself; we assumed the overall slope to be mechanically stable. We evaluated the slope stability by comparing the displacement value of a monitoring point at the end of cycling versus its value 1000 mechanical timesteps before. We described the slope as stable if the difference was smaller than 0.1 mm. In contrast, if it was unstable, we cycled 10.000 mechanical timesteps to check whether the model restabilized itself with some delay or decided that total slope failure occurred due to the unstable
model.



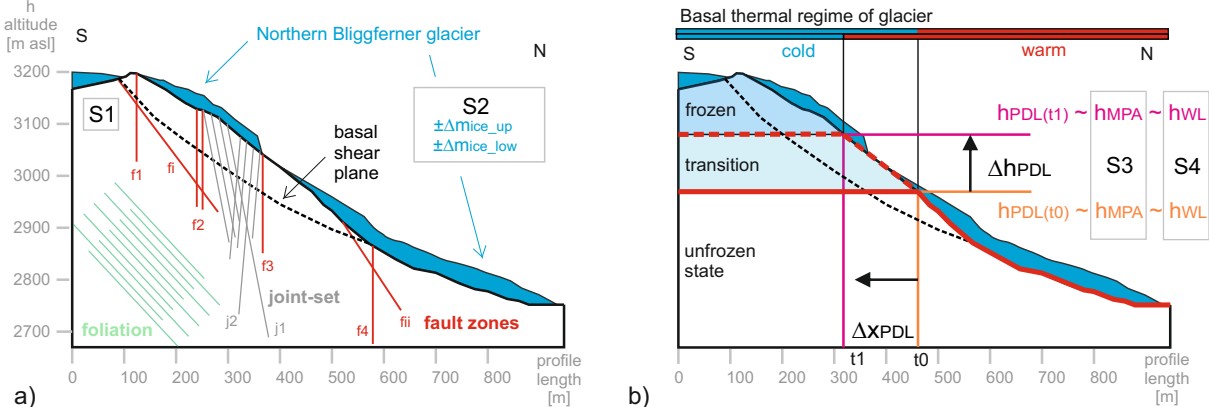

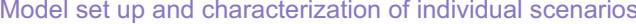

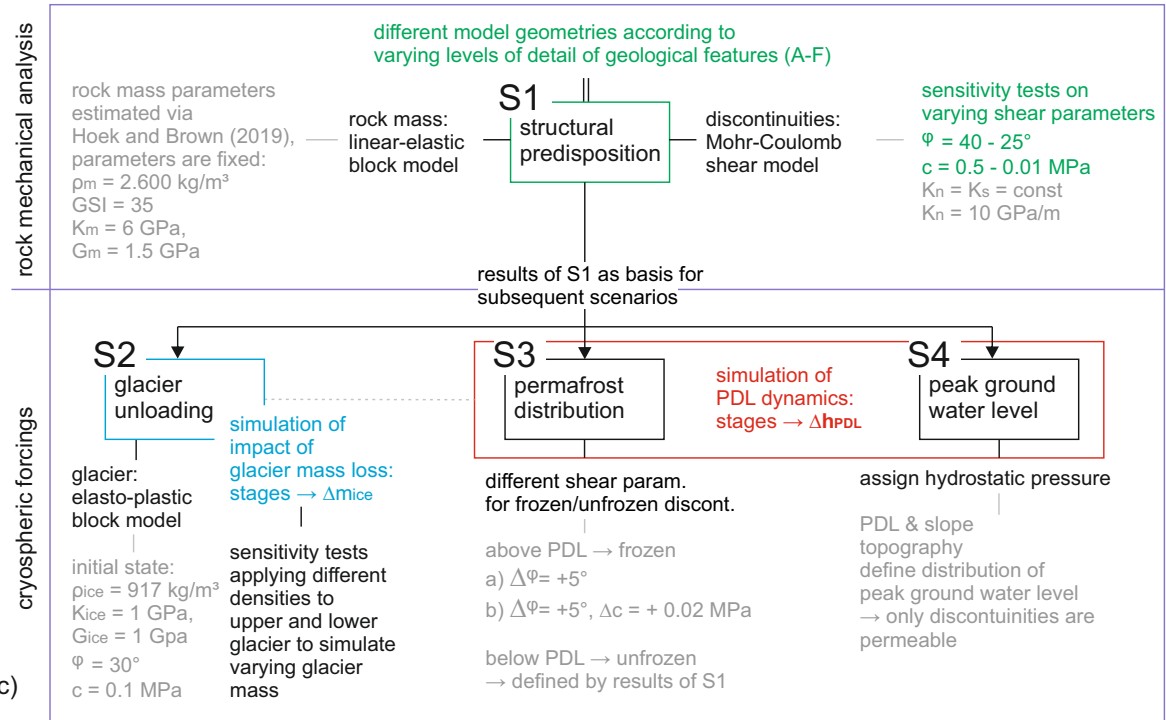

**Figure 3.** Conceptual scenarios for investigating rock slope mechanics and interdependencies with environmental forcings in a UDEC model. a) 2D cross-section for the Bliggspitze rock slide, including geological features and glacier extents of upper and lower sections of the Northern Bliggferner Glacier. b) The Polythermal Dividing Line (PDL) marks the border between cold and warm glacier bases in polythermal glaciers. The PDL poses implications on permafrost and water table within the rock mass: The elevation of PDL $h_{PDL}$ at its current state equals the Moutain Permafrost Altitude $h_{MPA}$ and the possible peak groundwater level $h_{WL}$. c) Description of the UDEC model setup and parameters for simulating the individual scenarios.



Changes in the cryosphere were simulated semi-dynamically by a stagewise model update. A stage represents a specific stadium of a glacier determined by the attributed load of the glacier in terms of mass $m_{ice}$ (S2), or a specific elevation of the PDL (S3 & S4). In the case of scenario S1, a stage represents a specific pair of shear parameters assigned to all discontinuities in the model domain. According to the selected scenario, we upgraded the stage after the end of cycling, and the decision on slope stability was made to compute for the next stage. With this approach, we iteratively simulated the mechanical response to observed shifts in cryopshere, such as a gradual increase of PDL to higher elevations.

## 4   Results

### 4.1   Pre-failure analysis of the Bliggspitze rock slide

#### 4.1.1   Evolution of the Northern Bliggferner Glacier

In the decades preceding the formation of the Bliggspitze rock slide, the Northern Bliggferner Glacier exhibited fluctuations in the rate of elevation change for the upper part, which appeared to neutralize the local effects of ice loss and gain (Fig. 4a). The lower part of the glacier exhibits a distinct loss in the range between 0 and $-1.3\ ma^{-1}$. In the period 2000-2012, covering the event of the rock slide, the upper part experienced a pronounced loss due to the slope deformation and the lower part gains in elevation by the deposits of the destabilized material by the Bliggspitze rock slide and other, smaller rock/ice avalanches during the respective observation period. The post-failure period indicates a process of glacier ice degradation in the upper part, while gain in elevation is observed in the less steep, lower part. The positive rate of elevation change observed at the glacier front in the is attributed to two main factors. Firstly, the deposits resulting from the ongoing activity of rock/ice falls/avalanches and debris flows, which partly cover the glacier ice, contribute to the gain in elevation. Secondly, the dynamics of ice flow transport glacier ice from the steeper to the shallower slope, where the conservation of ice is extended by the insulating effect of debris cover despite the relatively shallow altitudes. The glacier thickness models indicates a loss of $30\ \pm\ 5\ m$ in the lower part of the glacier between 1970 and 2003 (Fig. 4). In contrast, the upper part exhibited a less pronounced loss, with values generally below $10m$.

A comparison of historic orthophotography taken in the years of 1969 and 2003 from the upper section of the glacier where the head scarp is located (see Fig. 1) shows the evolution of the glacier by several features such as the opening of new crevasses, loss of ice aprons, loss of small glacier branch at side and drawdown of ice in proximity to the randkluft (Fig. 5a). A depression cone-like structure becomes evident in the right center of the 2003 orthophoto. The scarcity of snow cover and streamlines in the right part of both orthophotos are attributed to avalanches favored by the steep rock faces above the glacier. In 1969, the Bergschrund is well imprinted as a prominent transverese crevasse forming under extensive extensional stress (Fig. 5a, area 1). Further upglacier in an arch shape just below and following the ridge a narrower fractureband is visible. In the west, this linear fracture turns into a sequence of lateral crevasses. 34 years later (Fig. 5b), after the glacier surface steepened and lowered by about 10 m in the upper reaches, the crevasse pattern changed. The former bergschund is well visible but snow covered. Its location did hardly change. Moreover, a new gaping fracture has opened just above the old bergschrund. The fracture arch



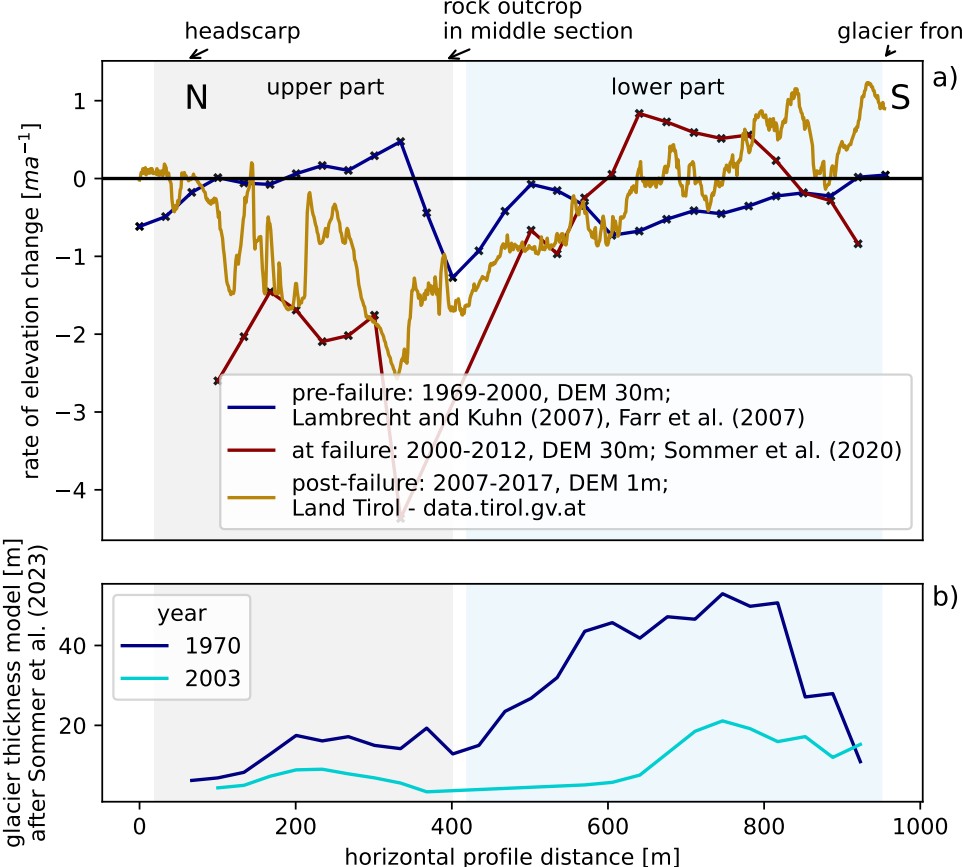

**Figure 4.** a) Rate of annual elevation change for given periods calculated along the longitudinal NS cross-section of the Northern Bliggferner Glacier, oriented in the direction of ice flow. b) Glacier thickness model according to Sommer et al. (2023) showing the pronounced loss in the lower part between 1969 and 2003. The trace of cross-section is displayed in Figure 1 and S4. The segmentation of the Northern Bliggferner Glacier into upper and lower part is shown in Figure 2a and 12.

further upstream is no longer a single line feature but disintegrated into many shorter extensional fractures some forming a crevasse sequence. All the new features are associated with increasing extensional stress.

### 4.1.2 Indication of permafrost in the area of the failure volume

The presence of permafrost in the area of the head scarp created by the rock slide is evident by the observation of ice-filled fractures and the ice aprons above the Bergschrund (Fig. 2, d). Permafrost conditions are further characterized by the analysis







**Figure 5.** Evolution of the upper part of the Northern Bliggferner Glacier by visual inspection of aerial photographs of the years 1969 and 2003 (data provided by Land Tirol - data.tirol.gv.at). Description of the features that changed during the 34-year interval: 1. Opening of new crevasses. 2. Loss of ice apron above the Bergschrund. 3. Loss of a small branch of the glacier. 4. A decrease in glacier thickness is visible through exposure to new rock exceeding several meters. 5. Loss of conjunction between Northern and Southern Bliggferner glacier.

of Ground Surface Temperature (GST), Electrical Resistivity Tomography (ERT) conducted on 12 August 2009, and electrical conductivity of mapped springs in 2011/2012 (for position see Figure 1).

From September 2013 to August 2016, 22 Ground Surface Temperature (GST) loggers were placed in the area of the head scarp within niches in the blocky terrain close to the topographic surface. All loggers showed mean temperatures of -4 to -1°C overall (Fig. 1). The upper / lower percentile of every GST record falls within a range of 0°C to -0.5°C / -3°C to -7.5°C (Fig. S2). Considering each hydrological year and GST record, the average number of annual Freezing Degree Days (mean GST below $0°C$) is 322. Most logger recordings showed a dependence of GST and incoming solar radiation: The lower the potential incoming solar radiation, the colder the mean annual temperatures (Fig. 1). The GSTs varied significantly in zero-curtain duration and winter air temperature signal penetration. From Sep 2014 to Aug 2015, extended zero-curtain periods were




evident, marked by daily standard deviations below 0.001°C. The preceding and following years showed fewer zero-curtain days and indicated less buffering of air temperature signals in autumn/winter/spring. Most GST records suggest snow-free conditions only in August and September, with daily standard deviations above 0.5 and GST closely following air temperature.

Figure 6 exemplifies the spread of heterogeneous records by drawing temperature for the coldest, a mid-temperature, and the warmest recorded mean of GST-logger. The record of logger S11 follows the trend of air temperature (AT) in a pronounced way and indicates snow-free windows or little snow cover throughout the year. In contrast, the location of the logger S12 is covered with snow for a minimum of 10 months per year. Nearly isothermal conditions are kept throughout winter and spring, with only light response to AT. Apart from S12, all other GST loggers resulted in maximum daily temperatures below $-3\ °C$

within the months of February and March, suggesting permafrost probable conditions according to Haeberli (1973).

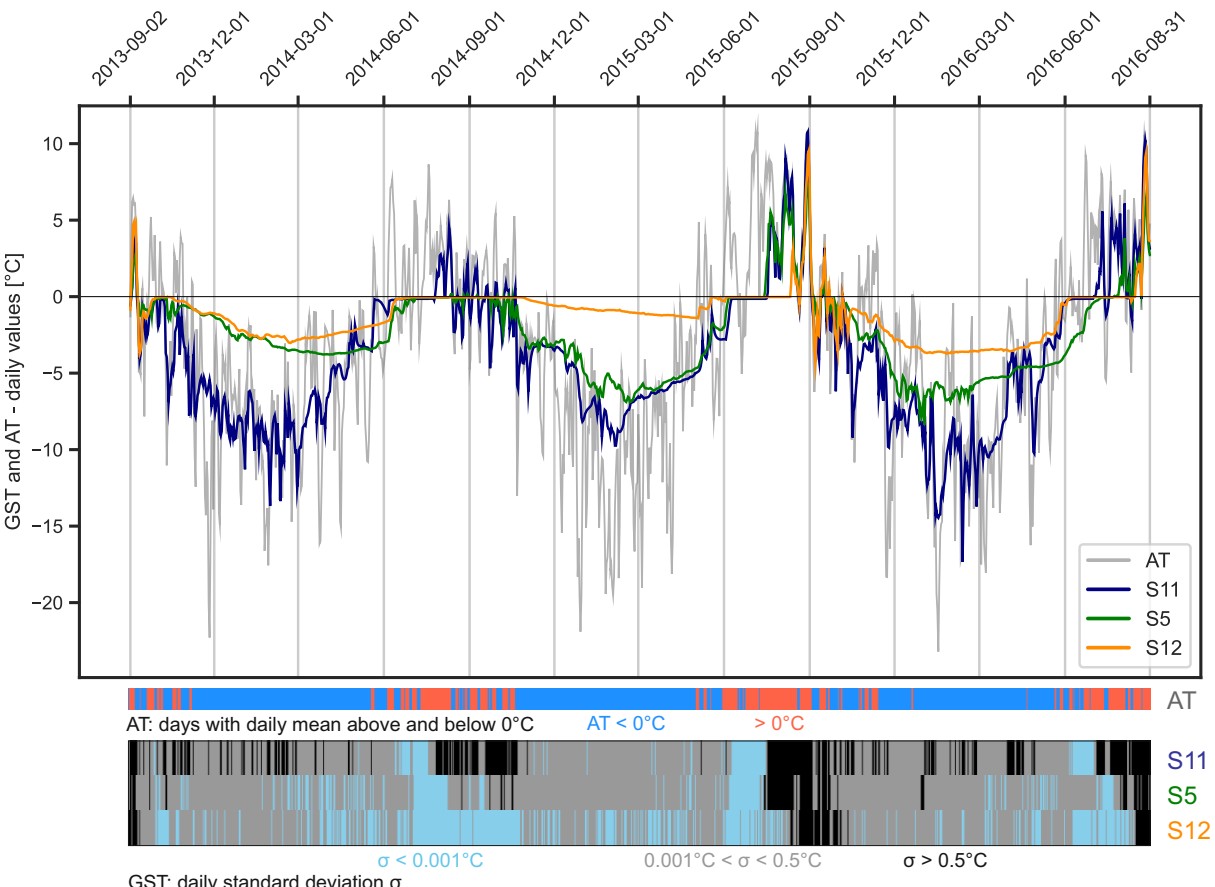

**Figure 6.** Ground Surface Temperature plotted for three distinct logger recordings showing strong heterogeneity in permafrost favorable conditions. Refer to Figure 1 for logger positions. The data was reanalyzed from data of Zangerl et al. (2019). A proxy for air temperature (AT) is modeled at the head scarp at an altitude of 3200 m asl: Difference of mean daily AT of two nearby meteo stations within a radius of 8 km (Pitztal glacier at 2863.9 masl and Brunnenkogel at 3437 m asl) was calculated daily and projected to the corresponding altitude using a linear model.



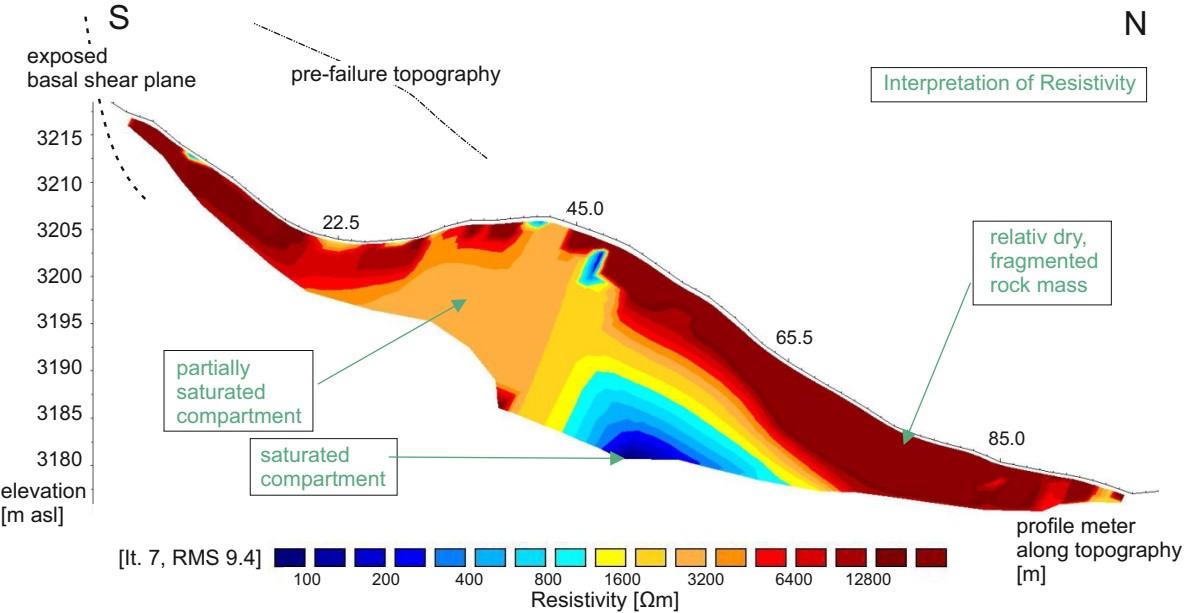

**Figure 7.** Inversion result of Electrical Resistivity Tomography conducted on the failed rock mass near the head scarp. The green-colored information in boxes illustrates the most probable interpretation of subsurface areas based on resistivity values as suggested in studies with similar lithology by Krautblatter and Hauck (2007); Keuschnig et al. (2017); Offer et al. (2024). For location of profile refer to Fig. 1

We carried out ERT on 12 August 2009, two years after the first time formation of the rock slide. The transect was located in the upper part of the fragmented rock mass that was covered with glacier ice before displacement. The transect starts at the contact of the fragmented rock mass to the basal shear zone following the direction of mass movement (see Fig. 1). The inversion result of the Electrical Resistivity Tomography (ERT) indicates the following subsurface situation (Fig. 7): At shallow

depths, high resistivities above $10k\ \Omega m$ are documented, most likely representing the fragmented, blocky rock mass with a high amount of air-filled voids that were clearly visible at the surface. The high density of contour intervals at profile meters 45 to 75 illustrates a decrease in resistivity by approaching higher depths. The low resistivities below $1k\ \Omega m$ indicate water-saturated compartments within the rock mass. Compartments in the mid-range of plotted resistivity values most likely represent a combination of voids within the rock mass and partially saturated areas. No clear evidence for frozen subsurface conditions

could be found due to resistivities generally below $20k\ \Omega m$. The interpretation of subsurface areas is based on resistivity values suggested by Krautblatter and Hauck (2007); Keuschnig et al. (2017); Offer et al. (2024)

We categorized the 32 springs according to the geomorphological forms in their vicinity and assigned the most probable source of the water origin (Fig. 8). A pronounced cluster of springs with high conductivity (> 150mS/cm), especially in proximity to the slopes surrounding the Northern Bliggferner glacier (see Fig. 1, which depicts the location of 18 out of 32

springs in the map), was observed. Additionally, two springs with high conductivity were attributed to rock glaciers, and one spring is directly linked to glacial discharge near the tongue of Northern Bliggferner Glacier. All other springs had electrical



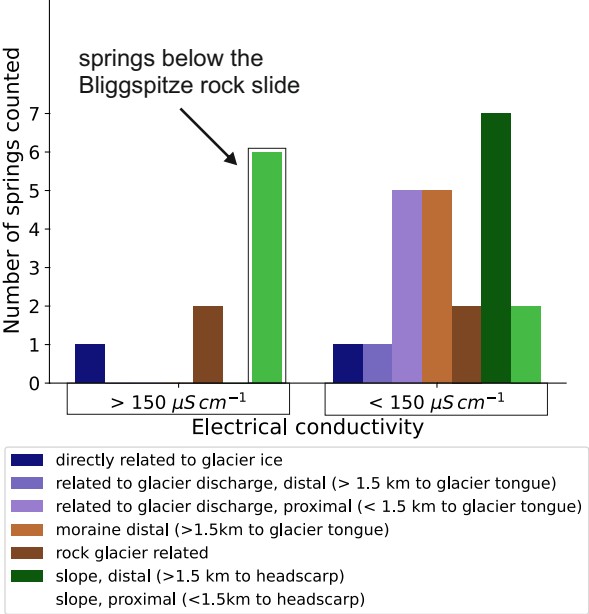

**Figure 8.** Categorical plot of 32 classified spring types concerning high or low measured electrical conductivity in the area surrounding the study site. 6 springs with high conductivity are located at the west-facing slope directly below the Bliggspitze rock slope. The reanalyzed data was measured in the summer of 2011/2012 (Dejean de la Bâtie, 2016).

conductivity values below $150\,\mu S\ cm^{-1}$. These included springs at different slopes at a greater distance to the head scarp, at moraines, at rock glaciers, and springs related to corridors of glacier discharge in a more distal area, as well as springs attributed to the catchment area of other glaciers.

### 4.1.3 Chronology and description of the formation of the rock slide

A comparison of the climate during the period before the first time formation of the rock slide with the long-term climatic conditions between 1780 and 2007 revealed that the period from January to June in 2007, before the rock slide in 2007, was the warmest in a 227-year climate record (Fig. 9). In 2007, the mean temperatures of the individual months January to March lay above the 75th percentile, April was the warmest month of the entire record, and May and June reached temperatures above the 75th percentile. Compared to the hot summer in 2003, winter and early spring also saw months with comparatively high average temperatures in 2007 (Fig. 9). Consequently, the period preceding the rock slide was characterized by elevated air temperatures for several months.

Before the formation the rock slide on 29 June 2007, several debris flows and rock falls had been noticed in the previous years. They confirmed this by examining photographs of the northwest-facing steep slope directly connected to the Northern Bliggferner Glacier. The activity of debris flows is evident by the erosion creating incised gullies (see Fig.1, change detection) and enlargement of talus in comparison to the state in 2003 (Fig. 2a, b). In the narrow time window days before the rock slide,



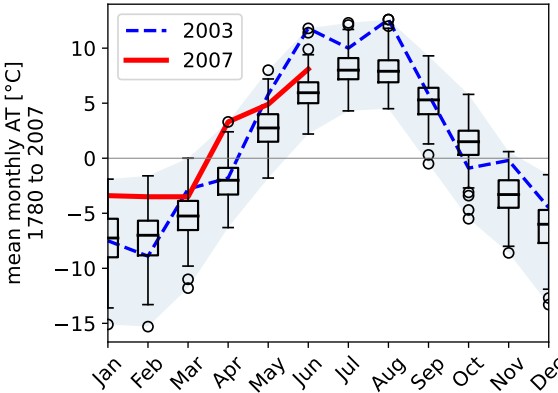

**Figure 9.** Climatic conditions at the study site during a 228-year record, including monthly temperatures from 1780 to 2007. The shaded area illustrates the range of historic air temperatures for the respective months. The red line illustrates the monthly temperature in 2007 prior to the formation of the rock slide (HISTALP dataset of corresponding 5-minute grid cell, GeoSphere Austria).

optical signs such as dislocation cracks in fresh snow, small magnitude rock and ice falls, and water leakages in slope and at rock outcrops were evident.

The rock slide was triggered at the end of a cold front that arrived four days earlier. The arrival of the cold front is marked by a daily precipitation sum of 20 mm, a drop in air temperature below 0°C, and a distinct drop in water temperatures of glacier discharge (Fig.10 a, b). The Vernagtbach meteo station, 8 km from the Bliggspitze site, served as a proxy for glacier and snow melt. From 17 to 26 June, discharge steadily increased, with electrical conductivity inversely related to this rise (Fig.10 b). The 25 June 2007 precipitation event is evident in the discharge record, with a tailing effect seen in the days following. The formation of the rock slide happened at a time that aligns with the daily peak in measured water temperature (Fig.10 b) at 11:23 am, as recorded by a video film.

Picture in Figure 2d) was captured hours after first time formation of the rock slide. The rock slide led to the fragmentation of the glacier ice and the exposure of previously ice-sealed bedrock (Fig. 2e). In the days after the event, springs within steeper slope sections close to the glacier were eye-catching. Orthophotographs recorded in the time interval between 2007-2009 and 2009-2012 indicate ongoing debris flow activity by streamlines and wet areas at the surfaces of the west-facing slopes below Northern Bliggferner Glacier (Fig. S3).

### 4.2 Laboratory experiments of anisotropic rock under frozen and unfrozen conditions

In addition to existing rock mechanical data related to permafrost dynamics, we investigated tensile strength as the most unknown and most sensitive parameter to thermal changes. We conducted destructive pseudo-Brazilian tests (BZT) on Paragneis rock samples to gain rock mechanical parameters simulating unfrozen conditions and frozen conditions. The difference between the two conditions is statistically significant (p-value / t-statistic of unpaired T-test: $5E^{-9}$, 7.10 (n=52)). We found that





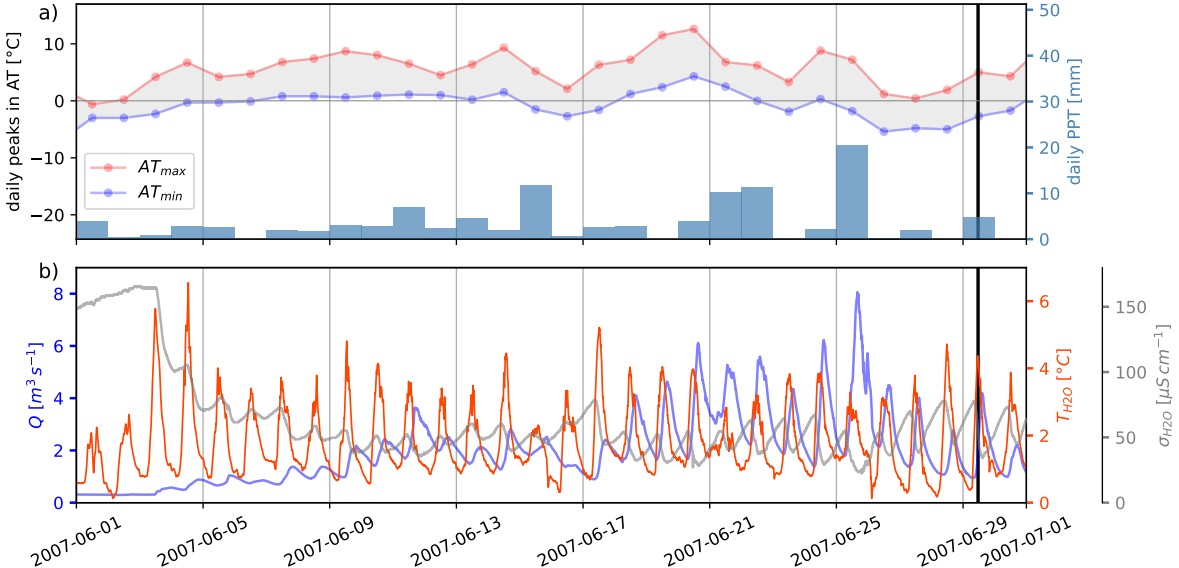

**Figure 10.** a) Weather conditions and b) meltwater discharge in the weeks before the rock slide occurring on 29 June 2007 at 11:23 am. The vertical black line marks the timing of failure. a) Daily peak air temperatures (AT) and total precipitation (PPT) for the grid cell of Bliggspitze field site (SPARTACUS dataset for corresponding 1x1 km grid cell, GeoSphere Austria). b) Hydrological record of the Vernagt basin: Water discharge (Q), water temperature ($T_{H_2O}$), and electrical conductivity ($\sigma_{H_2O}$) measured at 10-minute intervals. Location of the hydrological station: Oetztal Valley at 2640 m asl, 8 km air distance from the Bliggspitze field site (dataset published by Escher-Vetter et al. (2014)).

tensile $\sigma_t$ is higher for frozen than unfrozen states (Fig. 11). The mean value for $\sigma_{t,frozen}$ is 10.78 $MPa$ and for $\sigma_{t,unfrozen}$ is 6.50 $MPa$. The mean relative decrease in $\sigma_t$, transitioning from the frozen to the unfrozen state, is $-40\%$. Observing tensile strength as a function of the orientation of the foliation, a decrease in tensile strength transitioning from favorable to unfavorable orientation regarding force direction becomes apparent. Comparing one extreme case $\angle_{F_t/fo} = 0°$, with the other extreme
$\angle_{F_t/fo} = 90°$, a strength loss depending on the foliation orientation is evident. In extreme cases, the loss is more pronounced for frozen than for unfrozen states. However, cases representing the arbitrary orientation of foliation, such as those tested with $\angle_{F_t/fo} = 45$ or $70°$, do not indicate this trend. The surplus of tensile strength by freezing is more pronounced for favorable orientation $\angle_{F_t/fo} = 0°$ than for unfavorable orientation $\angle_{F_t/fo} = 90°$. In general, the transition from an unfrozen to a frozen state is more pronounced in tensile strength than the transition from an unfavourable to a favourable foliation orientation.

**4.3   Numerical discontinuum modeling of the Bliggspitze rock slide**

The mechanical modeling study was designed using multiple scenarios (S1 to S4, see fig. 3). In the initial scenario, we investigated the effect of structural predisposition, defined by the geology and the geometry of the rock slope (S1). In the second scenario, based on the results of the previous scenario S1, we analyzed the impact of cryospheric forcing on slope stability for three individual scenarios: S2 – glacier unloading, S3 – loss of permafrost, S4 – peak groundwater level.



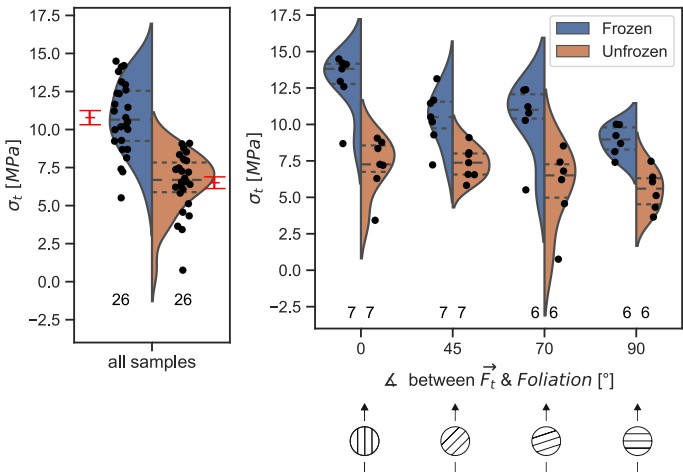

**Figure 11.** Tensile strength as a function of a) frozen and unfrozen states and b) of state and foliation of the paragneissic rock samples. a) shows the full pool of samples tested for frozen and unfrozen states. The red error bars indicate $\overline{x}$ and sd for the respective state. b) Categorisation of the samples according to tests conducted at different angles between loading force direction $\overrightarrow{F_t}$ and foliation orientation; see sketches below. The number of samples tested is given at the bottom of the graphs, ranging from 6 to 26.

The concept of the Polythermal Dividing Line (PDL) results from the investigations of the cryosphere, presented in the section *Pre-failure analysis*. It includes the consideration of the interplay between the glaciers' thermal regime, permafrost, and hydrogeology. The deviation of the concept is explained in the section *Methods* (see also Fig. **??**). The concept is applied in order to model the mechanical response of rock slope as the glacier transitions from a cold to a warm glacier ice regime while impacting permafrost distribution (S3) and hydrogeologic conditions (S4). The results of the simulations of individual

scenarios are presented first, followed by the results of the combined scenario simulations.

**S1 - Structural predisposition**: We examined 6 model geometries (A-F) with varying levels of detail of structural features to assess the structural control on the rock slope (Table 3). The degree of detail builds upon the previous level: fault zones only (A), plus joints (B), plus foliation (C). For the model versions D to F we included the basal shear zone and repeated the

previous sequence. As an example, the model geometry of E, including fault zones, joint sets and the basal shear zone, and the location of monitoring points that are used to evaluate the displacement of the rock slope are shown in Figure 12a. All tested model geometries and model states at the end of cycling can be visually compared in Figure S4 shown in the supplementary material. Subsequently, the results of the sensitivity study on structural features characterizing the rock slope are presented.

For all tested geometries, the inclined fault zone delineating the head scarp (Fig.3a, *fi*) showed the highest shear displacement

at states close to failure. For the models with geometry A to C, only stable states were reached. The geometry of intersected rock mass and the behavior of the linear elastic block models restrict the deformability of the rock mass, even at unrealistically low shear strength values. In case of geometry A to C, discontinuities do not dip out of the slope and the slope restabilized itself





**Table 3.** Set of model geometries used in scenario S1 for testing the structural control of the rock slope. The corresponding results are given for shear parameters of discontinuities of $\phi = 30°$ and $c = 0.1 MPa$ and for the model state after cycling 10.000 mechanical timesteps . Structural features marked with an asterisk are integrated in the corresponding model geometry.

| geometry | fault zones | joints | foliation | basal shear zone | state at end of cycling | max. displacement [m] within the model domain |
|---|---|---|---|---|---|---|
| A | * | | | | stable | 0.123 |
| B | * | * | | | stable | 0.099 |
| C | * | * | * | | stable | 0.116 |
| C.fo | * | * | * | | stable | 0.095 |
| D | * | | | * | stable | 0.130 |
| E | * | * | | * | stable | 0.122 |
| F | * | * | * | * | stable | 0.126 |
| F.fo | * | * | * | * | stable | 0.105 |

.fo .. discontinuities representing foliation structures are assigned higher values of friction angle: $\Delta\phi = +3°$.

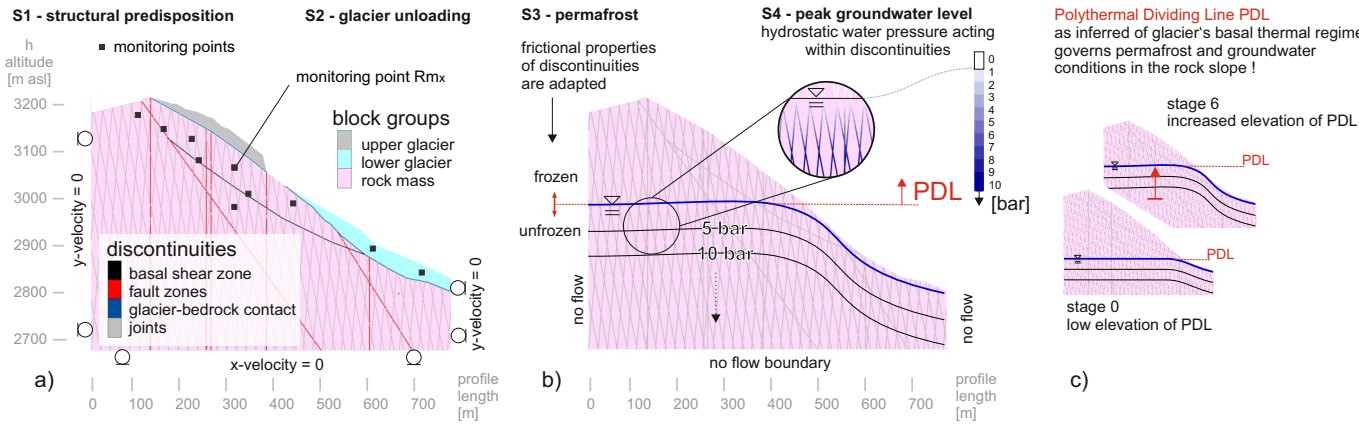

**Figure 12.** Implementation of model scenarios in UDEC. a) The geometry of model E shows the internal structural features, the dimension and shape of the glaciers covering the rock slope, and the location of monitoring points used to control displacement history while cycling. b) The Polythermal Dividing Line PDL is shown for an arbitrary elevation to indicate its effects on permafrost and peak groundwater level (S3, S4). c) Each stage in scenario S3 and S4 represents a certain elevation of PDL. With increasing stages, the elevation of PDL was increased. Note: The glaciers in a) were not included in simulations of scenarios S1, S3, and S4.

after initial displacement. Along vertical fault zones, upwards-facing scarps were created at various pairs of shear properties within the tested range ($\phi = 40 \ to \ 25, c = 0.5 \ to \ 0.01 MPa$). The offset of the resulting scarps was limited to a size of less than a centimeter. For geometry C, we included the foliation incorporated as a set of discontinuities and assigned a) the same shear strength value to all discontinuities or b) assigned slightly higher values to the foliation than for other discontinuities





($\Delta\phi = +3°$). For the case of a), overall shear displacement was accumulated over many neighboring foliation structures in the center of the slope, while in the case of b) shearing along foliation structures was hardly activated. By integrating the basal shear zone with geometry D, slope displacement could now propagate until complete slope failure occurred. Comparing results

of geometry D with geometry F, we observed larger total displacement within the full model domain when the model contained less structural features. A highly intersected rock mass containing more individual blocks due to a more detailed model geometry was found to be more susceptible to the interlocking of individual rotating blocks and could hinder further displacement. Figure 13 shows the result of sensitivity analysis for the different pairs of friction angle and cohesion for geometry E (fault zones, joints, and basal shear zone) by the history of the monitoring point $RM_x$, which is located at the center of gravity of

the displaced rock mass. Stable conditions of the rock slope can be achieved with either a relatively high friction angle and low cohesion or vice versa. All models with different geometries but the same shear properties of joints reached a stable state at the end of cycling. Due to high similarities at mechanic behavior and resulting maximum displacement, which varies in a range of $0.035\ m$, we decided to proceed with geometry E, representing the pre-failure state of the rock slope (Fig. 12). Unless otherwise specified, we fixed shear parameters of discontinuities to $\phi = 30°, c = 0.1 MPa$ for modeling subsequent scenarios.


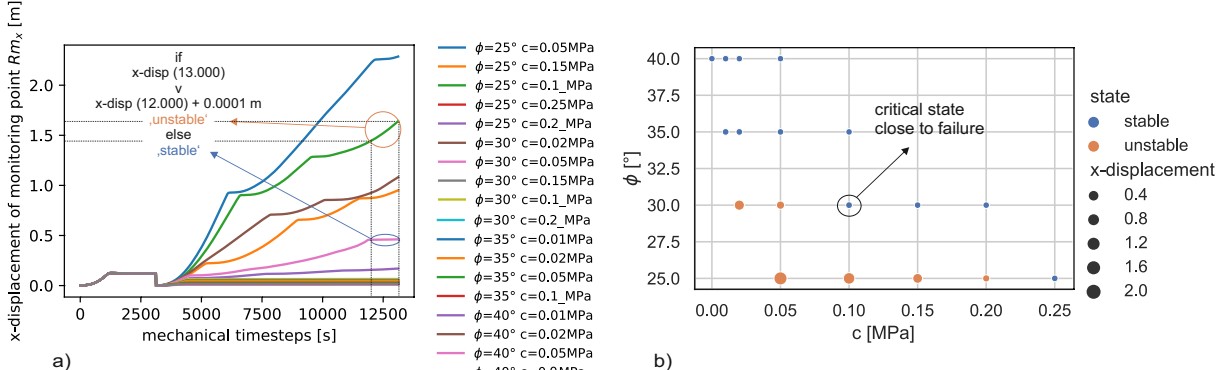

**Figure 13.** Sensitivity study of shear parameters, attributed to discontinuities in scenario S1 with geometry E, by analyzing the mechanic response by the history of montiroing point $Rm_x$. a) Relationship between mechanical timesteps versus x-displacement of the observed grid point for given pairs of shear parameters ("p25c0.05" equals $\phi = 25°, c = 0.05 MPa$). The given equation checks whether the model reaches equilibrium state at the end of cycling. b) Model states plotted for given pairs of shear parameters as a result of analysis of the x-displacement functions and maximum displacement in the x-direction at the end of cycling. The first 3000 mechanical timesteps show the cycling of the initialized model until reaching equilibrium.

**S2 - Glacier unloading**: We assigned varying densities to the geometrical model of the lower and upper part of the Northern Bliggferner Glacier (Fig.12a) and tested 9 different stages accounting for arbitrary glacier thickness. Figure 14 displays the maximum displacement of the monitoring point $Rm_x$ after the calculation of 10.000 mechanical timesteps according to different assumptions made about the glacier mass. All tested stages resulted in stable slope conditions with x-displacements varying

in a range from 0.04 to 0.08 $m$. We simulated gradual glacier mass loss by the comparison of stages with doubled ice density-



(7), with the assumed initial density- (2), with the halved density-glacier model (3), and with glacier-free conditions (8). The results showed ambiguous behavior: Higher normal load applied by glacier ice results in higher x-displacement of $0.045\ m$ (7), whereas a loss towards the assumed original stage (2) results in slightly lower x-displacement of $0.04\ m$. A further loss of glacier ice (3) does not indicate a distinct effect on x-displacement, whereas the ice-free stage (8) results in the highest

displacement of the four compared stages: $0.06\ m$. Stage 9 simulates the total loss of the lower part of the glacier, whereas the upper part remained unchanged, resulting in the highest of the observed x-displacements of $0.08\ m$.

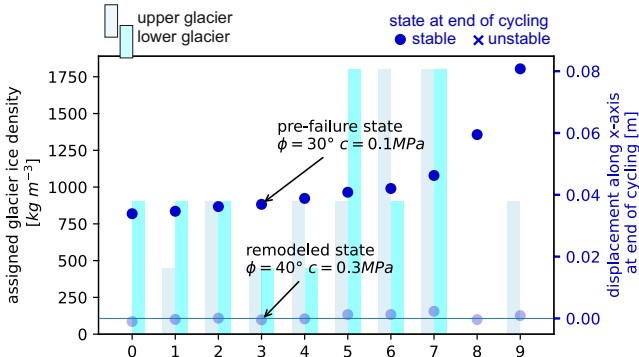

**Figure 14.** Scenario S2 simulates varying glacier mass's effect on slope stability. Results are illustrated for the end of cycling of 10.000 mechanical timesteps. Geometries of the upper and lower part of Bliggferner are illustrated in Figure 12a. The effect of mass unloading is simulated by the variation of assigned densities to the upper and lower parts of the Northern Bliggferner Glacier (see Fig. 3). Stage 2 displays the representative, initial model with an approximate thickness of 30-40 m for the lower and 20-30 m for the upper glacier with an attributed ice density of 917 kg/m$^3$. For example, stage 4 simulates a faster glacier mass loss at the lower part, whereas the upper part remains constant.

**S3 - Permafrost distribution**: According to the elevation of the Polythermal Dividing Line (PDL) of the Northern Bliggferner Glacier covering the rock slope, we divided the model into an upper, frozen, and lower, unfrozen part (Fig. 12b)and

assigned a higher shear strength to frozen states of discontinuities (for sub-scenario S3A: $\Delta\phi = +5°, \Delta c = 0 MPa$ and for S3B:$\Delta\phi = +5°, \Delta c = +0.02 MPa$) than for unfrozen states. By an iterative rise of the PDL, the history of monitoring point $Rm_x$ indicates displacements in the range of centimeters for low PDLs (Fig. 15). Simulating the retreat of permafrost with increasing PDL, the maximum x-displacement approached the magnitude of a decimeter, while the overall model remained stable for both S3A and S3B. Up to a specific elevation of PDL, a clear but constant difference in x-displacements was notable

between S3A and S3B within the same stages. However, when the PDL reached an elevation higher than 3100 m masl, meaning that the observed grid point $Rm_x$ located at 3080 m asl is now in the unfrozen area, the difference was equalized.

**S4 - Peak groundwater level**: Peak groundwater levels result from peak meltwater discharge or peak precipitation events. The temporary rise of the water table resulting from the event is delineated by the PDL and slope topography (Fig. 13b and

c). Water pressure is applied only within discontinuities according to the vertical offset to the defined water table. Simulating



the rise of peak groundwater level as a consequence of gradual increasing PDL, we found that no stage reached a stable state due to initiated continuous shearing along the basal shear zone (Fig. 15). When the groundwater level, defined by the elevation of the PDL, exceeds the height of the outcrop of the basal shear zone, irreversible displacement is initiated, and the area of the rock mass above the basal shear zone propagates downwards. Total slope failure occurs already at simulated low elevations of PDL due to the loss of the rock buttress at the toe: The exerted hydrostatic water pressure pushed out single blocks at the toe of the rock slope, and accelerated the mass above the basal shear zone that reached irreversible displacement (Fig. S5).

**Remodeling initial rock slope state**: As a consequence of the susceptibility of the slope to exerted hydrostatic pressure, we remodeled all scenarios with higher shear parameters of discontinuities ($\phi = 40°, c = 0.3 MPa$). These strength properties were determined by sensitivity analysis of scenario S4 under a fixed peak groundwater level at PDL of $3050\ m\ asl$, representing the least stable state prior to a further rise in water level to a PDL of $3080\ m\ asl$, which would result in total slope failure (for results see Fig. 15 & 14, marked with opaque dots). Our simulation results showed a limited impact of glacier unloading and permafrost degradation on slope displacement. While all conducted simulations resulted in stable states at the end of cycling - displacements were generally below $10^{-1}$ and $10^{-3}\ m$, respectively, for the case of modeled pre-failure state and the remodeled state. Simulations results of scenario S3A and S3B, given the shear parameters according to the remodeled state, display marginal x-displacements below $10^{-5}\ m$, which were not considered within a relevant range and therefore ignored in the plot of Figure 15. In contrast, applied water pressure within discontinuities, as simulated in scenario S4, led to total slope failure at the modeled pre-failure state when applied at the lowest parts of the basal shear zone only (displacements $> 1\ m$). At the remodeled state in S4, slope displacement increased continuously from $10^{-3}$ to $10^{-1}\ m$, concomitant with a rise of the groundwater table according to the PDL. The rock slope reached stable states at the end of cycling for elevations of PDL below $3050\ m\ asl$. Total slope failure with x-displacements $>\ 1\ m$ occured at elevations of the PDL above $3050\ m\ asl$, as it was defined by the back calculation of shear parameters for discontinuities (= remodeled state; Fig. 15).



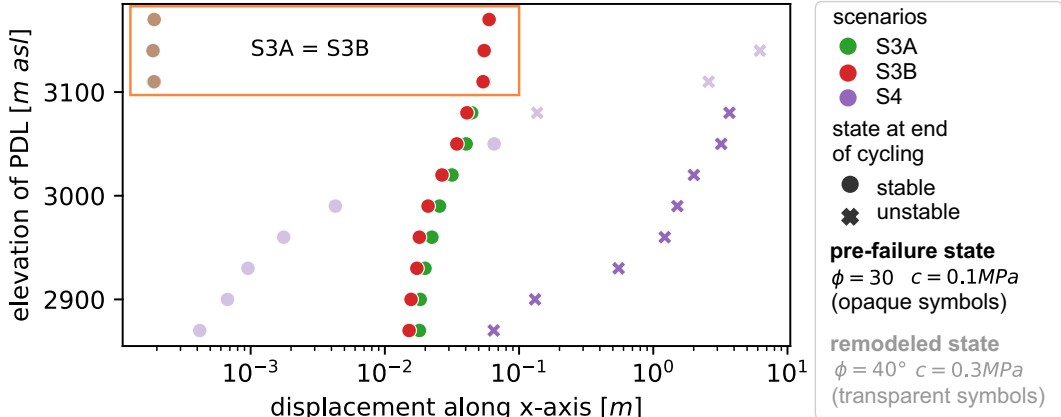

**Figure 15.** Semi-log plot: x-displacement of monitoring point $Rm_x$ after 10.000 mechanical timesteps for scenarios S3 and S4 verus the elevation of the Polythermal Dividing Line (PDL). The model geometry is according to S1E (see Figure 12), and parameters are assigned according to Figure **??**. Results are displayed for shear parameters of discontinuities given in the lower right of the graph. The values represented within the orange box demonstrate that, for the specified elevation of the PDL, there is no discernible difference in x-displacement between scenario S3A and S3B, for both assumed original conditions of discontinuities: the pre-failure and the remodeled state.

## 5 Discussion & Limitations

### 5.1 Interactions of glacier basal regime, hydrogeology & permafrost

For a long time, glacier research in the Alps and permafrost research in the Arctic lowlands were treated separately (Haeberli, 2005). The strong interactions between them and their relevance for geomorphological landscape evolution have been highlighted by Etzelmüller and Hagen (2005) conceptualizing thermal glacier regimes related to mountain permafrost. Among one of several documented, $> 10^6\,\mathrm{m}^3$ rock-ice avalanches, the Mt. Steller rock-ice avalanche (Alaska, 2005, release volume of $5\,(\pm 1)\cdot 10^7\,\mathrm{m}^3$) exemplifies how steep mountain glaciers in cold permafrost conditions induce complex thermal anomalies,

possibly increasing temperatures close to phase equilibrium at the glacier-ice-bedrock interface, which was hypothesized to have triggered the detachment by warming permafrost and meltwater infiltration at the base of the glacier (Huggel et al., 2008). However, due to the remote locations and restricted accessibility of detachment zones of glacier-permafrost-related rock slope failures, datasets and observations are scarce and typically challenging to obtain (Haeberli, 2005; Huggel et al., 2008; Shugar et al., 2021). From a glaciological perspective, glacier instabilities of type 2 are triggered by the transition of steep, cold hang-

ing glaciers into temperate regimes Faillettaz et al. (2015). Therefore, thermal modeling of the spatial and temporal evolution of the glacier bed temperature was emphasized in a first approach by Gilbert et al. (2015), who could clearly demonstrate a rise of the 0°C isotherm at the glacier-bedrock interface to higher elevations over multiple decades. The here presented publication delves below the ice-bedrock interface, analyzing glacier-permafrost interactions and their consequences on rock slope stability.



Results of the numerical modeling study could show a distinct impact of fluctuating groundwater levels on the stability of the
Bliggspitze rock slope. By employing the concept of a gradual rise of the cold/warm dividing line at the base of a polythermal
glacier (PDL), high amounts of available water in spring/summer may reach bedrock surfaces that were previously sealed with
cold glacier ice. The water then either discharges at the glacier base or infiltrates into the subsurface, resulting in temporary rise
of hydrostatic pressures within the fracture network. The Bliggspitze rock slide is likely to be prepared by transient hydrostatic
pressures, as shown by the simplified model scenarios, yet the processes behind are more complex and discussed subsequently
by utilizing Figure 16. Thinning of glacier ice under warming climate, and reduced magnitude and/or time of cold winter
temperatures penetrating the glacier ice, indicates that the polythermal Northern Bliggferner Glacier is shifting towards a
temperate regime. Conversely, glacier thinning under stagnant or cooling climate could also lead to the creation of extended
cold ice compartments Irvine-Fynn et al. (2011), which is not the case for the Northern Bliggferner Glacier as (a) long-term
trends in air temperature, (b) the accelerated loss of glacier ice visible at the lower part and the upper part in conjunction to
the Southern Bliggferner Glacier between 1969 and 2003, (c) debris flows originating from the rock slope below the glacier
reflect high water availability as observed in the years and days before the the first time formation of the rock slide and (d)
active layer deepening infered from the high mineralization of springs indicate the shift towards a temperate regime. Although
the altitude of 0°C isotherm of air temperature continuously increased since 1900, the full extent of the Northern Bliggferner
Glacier remained considerably above the 0°C isotherm (HISTALP dataset, Chimani et al. (2013)). Thus, diffusive heat flow
alone cannot describe the observed warming of the glacier, and advective heat transport must be considered as relevant process
for warming glacier ice. Considering the rather cold climate, the aspect and slope of the terrain, and the stagnant glacier front
between 1969 and 2006 (Fischer et al., 2015), the movement of the glacier must be marginal. Ground Surface Temperatures
generally reveal permafrost favourable conditions at the elevation of the head scarp conducted in the years after the failure. In
contrast, the distinct loss of ice aprons above the Bergschrund of the Northern Bliggferner Glacier failure visibly demonstrates
permafrost degradation in the decades before the failure.

In our generic model, we assumed (i) the existence of a single Polythermal Dividing Line at the bedrock-glacier contact
(PDL), (ii) that the Mountain Permafrost Altitude (MPA) is bound to the elevation of the PDL that is gradually rising under
a warming climate, (iii) that the area below the PDL is unfrozen and discontinuities below the water table, which aligns with
PDL and slope topography, are fully saturated. These idealistic perspectives are discussed below.

Firstly, the existence of cold basal glacier ice is not only dependent on the altitude and might be distributed discontinuously,
especially in complex topographic terrain and heterogeneous flow velocities (Cuffey and Paterson, 2010). We differentiate
between warm and cold basal ice by temperature and basal discharge, which is only possible under warm ice at the base. In
the case of the Northern Bliggferner Glacier, we observed the loss of the conjunction to the Southern Bliggferner Glacier in
the period 1969 to 2003. This led to the exposure of a ridge with S-oriented rock slabs to direct solar radiation ($Q_{rad.}$) and
to an accelerated conductive warming of rock. Shortly before the failure, basal warm ice is likely to be found to a greater
extent at the lower altitudes and to a smaller extent at the higher altitudes in the area of the S-exposed ridge and the Southern
Bliggferner Glacier. Basal cold ice is most probable in the upper part of the Northern Bliggferner Glacier likely to be framed in
between warm ice compartments in the years short before failure. Near-surface permafrost indicates cold ice compartments in





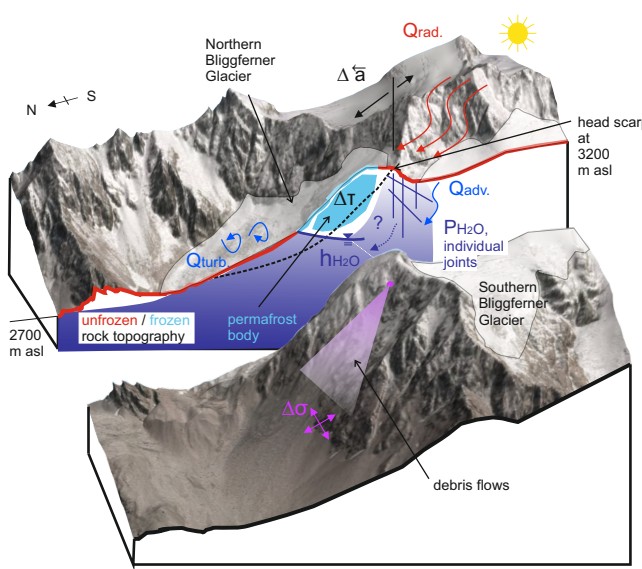

**Figure 16.** Holistic view on the manifold processes leading to the Bliggspitze rock slope failure. For dicussion of individual concepts and abbreviation refer to the text.

proximity and is evident in the N-oriented rock faces above 3200 m in proximity to the Northern Bliggferner Glacier, as shown by GST and field observations, and likely to be absent in the area of the Southern Bliggferner Glacier due to the exposition of the glacier to solar radiation and infiltrating meltwaters from the steep west facing couloirs above.

Secondly, the distribution of mountain permafrost with low ice content strongly depends on altitude, aspect, slope, terrain roughness, duration and thickness of insulating snow/glacier ice/vegetation cover and on the inherited thermal signals of large rock mass storing the transient temperature effects of Holocene time scale Noetzli and Gruber (2009). More than 80% of the failed rock mass was thermally insulated by glacier ice before the failure, whereas the steep surface of the W-facing mountain flank underneath allowed undampened air temperature signals to penetrate the rock. Owning the three-dimensional situation, permafrost is likely to be irregularly shaped due to the complex topography along with heterogeneity in snow/ice cover resulting in varying ground surface temperatures (Haberkorn et al., 2015). Nevertheless, the warming of air temperatures at the study site since the Little Ice Age, and especially in the last decades, is likely to have driven the lower base of the mountain permafrost to higher elevations by conductive heat transport (Biskaborn et al., 2019). In addition, the presence of abundant surface water strongly modifies the shape of permafrost. Advective heat, transported by percolating waters, is an effective mechanism for degrading clef-ice and creating thaw corridors in fractured rock mass (Gruber and Haeberli, 2007; Hasler et al., 2011). In fractured aquifer systems, water flows primarily along fractures, while the intact rock matrix exhibits



minimal hydraulic conductivity ($K$ for low porosity rock is generally in the range of $10^{-8} - 10^{-10}\ ms^{-1}$). Linked thaw corridors create wedges that induce bottom-up permafrost degradation under saturated conditions and lead to accelerated deepening of permafrost (Magnin and Josnin, 2021). Moreover, rainwater or meltwater heated by the flow over sun-exposed rock slabs are sources of water with temperatures above $0°C$ contributing to an accelerated cleft-ice degradation ($Q_{adv.}$). The ridge, forming the headscarp, and the west-facing rock couloirs above the Southern Bliggferner Glacier exhibit conditions that

favor the infiltrating of such pre-heated water at the elevation of the detachment zone. The temperature of glacier or snow meltwater is typically $0°C$, considering short travel times. In this case, advective heat transport is less effective for cleft-ice degradation, as part of the water refreezes while infiltrating into the ground. The small gradient between water and ice/rock temperature substantially delays the process of the degradation of ice (Cuffey and Paterson, 2010). It is notable to mention that both, hydrostatic water pressure and solute concentration, are not very effective in lowering the melting point for the purpose of

cleft-ice degradation in alpine environments, due to the high water pressure (equal to $1000\ m$ in hydraulic head) or high water molality ($0.54\ mol/kg$) that are required in order to lower the melting point by $-1°C$ in temperature (Krautblatter, 2010).

   Thirdly, the system feedback of basal glacier warming leading to meltwater production and subsequent loss in permafrost underneath incorporates a highly simplified view in our model. Permafrost might outlast several decades or centuries when being fed with cold water around $0°C$. The permeability of fractured rock slopes affected by permafrost is mainly influenced by

the properties of the fractured rock aquifer - fracture density, persistence, aperture and fracture infills, and the interconnectivity of fracture system (Hakami, 1995) - and by the presence of ice-sealed discontinuities being features of permafrost (Woo, 2012). The permeability of ice-sealed fractures in granite was found to be one to three magnitudes lower in comparison to the thawed state (Pogrebiskiy et al., 1977). The spatial contrast in permeability, i.e. due to (i) the cluster of ice-sealed and ice-free fractures (ii) varying system connectivity, fracture apertures, and geometries of geological structures such as bedding/foliation,

joints, fault zones or (iii) strong gradient of weathering that reduces with higher depths in combination with high surficial water discharge might lead to the build-up of localized hydrostatic water pressures ($P_{H_2O}$ at individual fractures). Hydraulic heads of several decameters were derived for individual fractures within permafrost rock at Zugspitze (GER). While extreme precipitation events led to hydraulic heads of $40 \pm 10\ m$, average daily snowmelt resulted in $27 \pm 6\ m$ Scandroglio et al. (2024).

   Despite the rather slow process of permafrost degradation, given infiltrating meltwater of $0°C$, the permeability of rock

mass, including ice-sealed discontinuities, can be increased by other mechanisms, which are described below.

     – Prefailure activity, as observed at many high-volume rock slope instabilities in permafrost Caduff et al. (2021); Hilger et al. (2021); Etzelmüller et al. (2022), is indicated by continuous or accelerated movements of parts within the susceptible rock slope or even the entire slope and may affect local hydrogeology by the creation of new fractures and voids. As a result of glacial thinning and the observed debris flows in the faces underneath the Bliggspitze rock slope, the local

stress field changes in response to glacier or rock debuttressing ($\Delta\sigma$) and may therefore create new fractures or initialize small movements indicating pre-failure activity.

     – Water pressure can mechanically widen fracture aperture, challenging the tensile strength of potential ice infillings, and enabling localized channeling of water to higher depths by changes in hydraulic aperture. According to the cubic law for



fluid flow, hydraulic conductivity is proportional to the cube of the fracture aperture (Witherspoon et al., 1980), meaning small changes in aperture significantly affect hydraulic conductivity. Hydraulic packer tests conducted in a sparsely fractured granite within boreholes at the underground research tunnel of the Korea Atomic Energy Research Institute (KAERI) could show a distinct increase in fracture aperture at relatively low hydraulic pressure: An applied hydraulic head of $20\ m$ / $50\ m$ resulted in a change of fracture aperture by a factor of 1.22 / 1.44 on average (Ji et al., 2013). Hydrostatic water pressure influences effective normal stresses, which increase the permeability of fractures due to the non-linear closure behavior of fracture aperture, especially under low normal stresses Zangerl et al. (2008). Alternatively, shear dilatation, likely to be favored by the influence of hydrostatic water pressure, increases permeability by changing the channel geometry, concentrating the flow path along concerned fractures (Min et al., 2004).

There are certain indications that pressurized, cold water encountering ice-infillings in joints under turbulent water flow ($Q_{turb.}$) might create scours in the ice by mechanic erosion due to the contact with water vortex' (similar to the creation of moulins within the glacier). Viscous dissipation from turbulence generates heat, increasing temperature gradients and potentially accelerating ice degradation. Turbulency, influenced by flow velocity and stream geometry, is more likely to occur in highly connected, fractured rock masses exposed to periglacial weathering and under high water pressure resulting from glacier discharge. These effects may be limited to shallow depths and are challenging to quantify in natural conditions. The impact of turbulence on cleft ice degradation and the subsequent increase in permeability is believed to be relatively insignificant when compared to the previously discussed mechanism.

Considering glaciers' basal water discharge in spring/summer, a temporary rise of the water table is likely to occur up to the elevation of the PDL. Although permafrost may last longer given the discharge of glacial meltwater, it is most likely that water will infiltrate to greater depths along preferred high permeability pathways, whether or not permafrost is present. Still, hydrogeology in fractured permafrost rock mass is scarcely researched and explained mainly by conceptual considerations rather than by real in-situ observations.

## 5.2 Mechanical analysis of Bliggspitze rock slope failure and implications of results

### 5.2.1 Structural predisposition and limitations of the modeling framework

Rock slope failures in brittle fractured rock are governed by shearing along pre-defined discontinuities. As the UDEC framework does not allow for arbitrary crack propagation through intact blocks and enables shearing only along explicitly defined discontinuities, all geometrical structures are integrated as fully persistent structures. Rock bridges are considered by the cohesive term within the Mohr-Coulomb shear criterion that was assigned to all discontinuities (see comparable rock mechanic studies i.e. Fischer et al. (2010); Gischig et al. (2011); Mamot et al. (2021)). Unlimited slope displacement, indicating total rock slope failure, could only be represented by explicitly modeling discontinuities that are dipping out of the slope and allowing for kinematic freedom of the blocks. Therefore, the integration of the basal shear zone was essential to enable total failure in the UDEC simulations.





Given the geologically predisposed state of a rock slope, altered by hydro-thermo-mechanical forcings over multi-glacial cycles as described by Grämiger et al. (2018, 2020), the here conducted mechanical modeling study emphasizes the analysis of the final triggering event leading to the failure and the formation of a fully peristent shear zone. In order to generate a model representing the pre-failure state of the Bliggspitze rock slope in the decades before the failure, we ran sensitivity tests on shear parameters of discontinuities and structural geometry to determine a critical but stable state (see scenario S1). The hypothesized pre-failure state can be described by various combinations of friction angle and cohesion. It can be interpreted in terms of characteristics of the basal shear zone, as this structural feature had the main control on rock slope displacement. The chosen pair of shear parameters of $\phi = 30, c = 0.1 MPa$ inidcate an overall low shear strength of the rupture surface, which is possibly attributed to wheatered/alterated material, foliation orientation, and tectonically stressed and deformed structures within the gneissic rock mass. For a reference, the low friction angle $\phi$ is slightly higher than tested residual friction angles of brittle fault zone material $\phi_{res.} = 25.7 \, to \, 28.9°$ (Strauhal et al., 2017), while the cohesion c leads to the conclusion of a low proportion of intact rock bridges.

We argue that the basal shear zone follows a sequence of steeply inclined fault zones and aligns with the orientation of articulated foliation in the lower part of the slope. The modeled foliation does not impact the slope instability but was modeled only by straight-line structures (see scenario S1 geometry C). As observed at the field site, the foliation of rock mass follows the southern limb of the NS-oriented tectonic synform at the concerned rock slope and is slightly bent. On the opposite side of the valley from the Bliggspitze rock slope, morphology exhibits extended rock slabs following the tectonic structure of the Northern limb of the synform. The tectonic setting suggests that discontinuities following the foliation pattern dip out at the lower toe of the slope favoring the development of continuous and curved basal shear zone.

### 5.2.2 Mechanical response to a changing cryosphere

**S2 - Glacier unloading**: Galcial unloading indicates limited impact on rock slope displacement varying in the range of $0.03 \, to \, 0.08 \, m$, as modeled for the Bliggspitze rock slope. The glacier was modeled as an elastoplastic material creating the least possible shear resistance to the adjacent intruding rock mass. This was consistent with findings of McColl and Davies (2013); Grämiger et al. (2017), who concluded that a glacier acts as poor buttress due to the ductile behavior of ice under small strain. We divided the glacier into an upper and a lower part. The results proved the mechanical logic that load, applied to the upper part of the rock slope only, leads to the highest simulated displacement (Fig. 5: stage 9), whereas load, applied to the lower part only, leads to the lowest displacement (stage 0 - buttress effect).

Modeling both the upper and lower part of the glacier, the results indicate mixed effects attributed to (i) unloading and (ii) to inertia/selected material model of the lower glacier covering the outcrop of the basal shear zone and affecting unhindered sliding of the failing rock slope. The latter is an inherent effect of UDEC mechanics, imposed by the approach of modeling the glacier under a stationary geometry with elastoplastic material. As the focus is on rock slope mechanics rather than glacier ice deformation, the applied material properties were limited to a lower boundary in order to prevent the glacier from freely deforming under its own weight, while concentrating on displacements within the rock mass. In contrast, other UDEC studies with a similar modeling approach have shown a pronounced effect on rock slope displacement for retreating valley glaciers



Fischer et al. (2010); Rechberger and Zangerl (2022). They differ from the Bliggspitze study site by considering a higher change in glacier thickness of more than $100\,m$, and both of them have in common that slope failure occurred when the slope was almost ice-free.

Despite the mentioned handicap in the modeling approach and the lack of recreating the results of comparable studies, we hypothesize that mass loss of the Northern Bliggferner Glacier was one of several factors preparing the rock slide: Firstly,
the Northern Bliggferner Glacier has lost several decameters of ice in the lower part while remaining stagnant in the upper between 1970 and 2003 (Sommer et al. (2023), see Fig. 5). Secondly, the relatively rapid changes in the stress field lead to stress redistribution and are likely to affect crack propagation and failure initiation, including the failure of rock bridges, by sub-critical fracture propagation (Atkinson, 1982).

**S3 - Permafrost distribution**: By iteratively incrementing the mountain permafrost altitude, the resulting rock slope dis-
placement increased from a minimum of $0.1\,m$ to a maximum of $0.08\,m$. We applied a highly simplified permafrost model by (i) distinguishing only between frozen and unfrozen areas, (ii) assuming uniform distribution delineating unfrozen/frozen areas by a horizontal line, (iii) neglecting active layer dynamics and (iv) accounting for permafrost by increasing the shear strength of all discontinuities within the frozen area by synthetic values that derived from theoretical considerations on rock-ice mechanics as proposed by Krautblatter et al. (2013).

To date, the influence of permafrost on the mechanics of deep-seated shear zones located at depths $> 30\,m$, as in the case of the Bliggspitze rock slope, is poorly understood. Conversely to shallow shear planes, where ice-infillings control shear strength dependent on the thermal state Mamot et al. (2018, 2020), rock-rock contacts of joints may suppress the effect of ice-infillings at higher normal load. Moreover, low deformation rates favor the creep of ice which does not affect overall shear strength under low strain (Hobbs, 2010). At deep-seated shear planes and zones under warming permafrost, the effect of ice-infillings
on overall shear strength might be minor, but mechanical degradation of intact rock bridges become more relevant, as intact rock properties are affect by rising temperature too Mellor (1973); Inada and Yokota (1984); Draebing and Krautblatter (2012).

In scenario S3A, we assigned a surplus in friction angle $\Delta\phi = +5°$ to frozen discontinuities derived by the results on frozen/unfrozen shear tests at normal stress in the range of $0.6\,to\,1.3MPa$, equivalent to depths of the shear plane of $25\,to\,50\,m$ (Krautblatter et al., 2013). In scenario S3B, we additionally considered the mechanical degradation of rock bridges by attempt-
ing to incorporate insights from the conducted tensile strength tests under frozen/unfrozen states. However, the actual distribution and quantity of rock bridges is unknown, leading to a blurred view on how to translate the results into the Mohr-Coulomb shear as suggested by the concept of Kemeny (2003). Further research and verification of the model utilized in this study are essential for a comprehensive understanding of the mechanics of deep-seated shear planes in permafrost.

**S4 - Peak groundwater level**: The modeled pre-failure state of the Bliggspitze rock slope proved to be hypersensitive to
changes in groundwater level. Total failure occurred when the lower outcrop of the basal shear zone was exceeded by the groundwater table. In addition, we used a reverse modeling approach, asking what shear parameters would allow the model to remain stable given a defined peak groundwater level at an elevation of 3050 m asl which corresponds to the center of gravity of the failing rock mass: In order to ensure stable conditions, the shear properties must increase by $\Delta\phi = +10°\Delta c = +0.2MPa$ in comparison to the pre-failure state, suggesting a pronounced destabilizing effect of hydrostatic water pressure on slope stability,



which was also reported by other UDEC studies in periglacial environments (Fischer et al., 2010; Stoll, 2020). For the given reverse modeling state, a hydrostatic water pressure of approximately $0.5\ MPa$ acted on the basal shear zone, counteracted the weight of the rock mass above, and asserted pressure gradients on joint walls within inclined/vertical discontinuities.

In the case of the Bliggspitze rock slope, peak groundwater levels can result from transient peak discharge of meltwater or intense liquid precipitation (Magnusson et al., 2014) or abrupt release from englacial stored water (Fountain and Walder, 1998).

Although the hydraulic conductivity of the rock slope was not quantified, we assumed fully saturated conditions to exist due to the imbalance of high peak discharge versus an assumed limited infiltration capacity. The state of transient peak groundwater level was simulated by a static, unconfined water table, defined by the elevation of PDL and the bedrock's surface topography below the warm glacier base (Fig. 12). In contrast to Grämiger et al. (2020), who stated that basal water pressure could reach values close to the pressure of ice overburden, we neglected the effect of subglacial water pressure at the base of the glacier

due to the small thickness of the overlying glacier and the geometry of slope favoring surficial run off at the glacier base.

### 5.3   Permafrost characterization and interpretation

GST records display heterogeneity that is most likely attributed to differences in locations of measurements accounting for the variability in snow depth and annual duration of snow cover (daily standard deviation of GST measurement as interpreted by (Haberkorn et al. (2015); Draebing et al. (2022), Fig. 6), and for the microscale roughness of topography and aspect (see

Fig. 1). The surface of the rock slope is characterized by blocky, heavily weathered rock, which is distributed in a debris-like soil layer over the fractured rock mass below. This layer contains a high proportion of voids. Therefore, direct solar radiation may be less efficient at warming the rock mass because cold air trapped in voids acts as addional buffer. The measured GST, elevation, aspect, and the high content of voids collectively support the favourable conditions for the existance of permafrost in the area of the head scarp of Bliggspitze rock slope failure.

The ERT measurement conducted on the fragmented rock mass two years after failure indicates permafrost-free conditions and water-saturated areas in the lower part of the transect according to the interpretation logic applied in other ERT-studies (Krautblatter and Hauck, 2007; Keuschnig et al., 2017; Offer et al., 2024). The unusual finding is, that if the glacier covering the rock mass was already warm-based before failure, meltwater infiltration might have eroded the preexisting permafrost under cold-based conditions (the mechanism was discussed above). If permafrost was still there upon failure, frictional energy

dissipation caused by rock slide and fragmentation of rock mass (Erismann and Abele, 2001) might have eroded permafrost within the area of affected mass movement.

Water outlets' measured high electrical conductivity values at nearby springs could be used as an indicator for active layer deepening (Colombo et al., 2018) or ongoing glacier erosion (Collins, 1979) depending on the source the water is stemming from. In surroundings to the Bliggspitze rock slide, groundwater mineralization can be influenced by landslide activity (Bo-

gaard et al., 2007). Still, it is unlikely that the higher measured conductivity is attributed to tailing effects due to rock slope failure that occurred 4 years before. If recent landslide activity, glacier or rock glacier as a source for mineralized water could be excluded, we inferred the water source to be permafrost. Therefore, we interpret the observed dense cluster of springs with high electrical conductivity below the Bliggspitze rock slope as active layer deepening and loss of permafrost.





### 5.4 Implications of frozen rocks on mechanics of deep-seated shear planes and anisotropic rock fabric

The laboratory experiments are designed to investigate tensile strength for frozen/unfrozen states of brittle intact rock (to account for mechanical degradation of rock bridges under thawing permafrost) and to analyze the structural control of the foliation.

Rock bridges can be incorporated into the Mohr-Coulomb shear criterion through the use of the cohesive term, represented by the parameter of fracture toughness mode II and geometry (Kemeny, 2003). As the results of the Brazilian test are linearly
related to fracture toughness mode II (Hua et al., 2017), and require less expensive sample preparation, we have elected to utilize the Brazilian test. The results of the conducted BZTs clearly demonstrate higher values for frozen than unfrozen rock samples. Consequently, we argue that even for deep-seated shear planes, permafrost can increase overall shear strength by increasing the shear strength of individual rock bridges, irrespective of the presence of ice infillings.

The results of the BZT do not indicate that the foliation of intact rock impacts is structurally controlling the rock slope
mechanics. The two extreme cases (parallel and perpendicular sample arrangement) indicate dependence on foliation, which is more pronounced for the frozen than for the unfrozen sample group. Given the limited number of samples tested (n=6 or 7), the observed variation within each case, which was at least 4.5 MPa, The variation is primarily attributed to the natural heterogeneity of the samples. Additionally, the observation that fracturing occurred along the defined area spanned by the opposing load bridges, regardless of the orientation of foliation, indicates that the tested Paragneiss can be described as an
rather isotropic material. Apart from the existing foliation-parallel joints, which define distinct shear planes, test results suggest that anisotropy of rock fabric in intact rock is mechanically irrelevant for the Bliggspitze rock slope failure.

The increase of $\sigma_t$ from an unfrozen state at T = +20°C to a frozen state at (-10°C) of +65.8% is distinctly higher than the findings of comparable studies. Mellor (1973) reported an increase from unfrozen (+23°C) to frozen state (-8°C) of approximately +8.4% for saturated granite. Inada and Yokota (1984) reported an increase from unfrozen ($T_{room}$) to frozen state
(-40°C) of +70.1% for saturated granite. In contrast to commonly applied constant strain, we manually simulated constant stress tests and reduced the time to perform a single test. We hypothize that the quick increase in applied load restricts the possibility of ice relaxation/degradation and sample deformation before reaching peak tensile strength. Thus the performed tests results in higher tensile strength values compared to previous studies.

It is of interest to note that the test results pertain to areas within the geological unit that exhibit greater strength and are more
erosion-resistant. When collecting samples, besides homogeneity and lithology, accessibility and transportation were criteria for choosing a site. Samples were taken from block streams with low content of fine grains, indicating periglacial phenomena in a relict state Oliva et al. (2023). In these circumstances, blocks found on top of the block stream today represent the most weathering-resistant parts.

### 5.5 Synthetic discussion

The Bliggspitze rock slide is an example of a hitherto underexplored but commonly occurring phenomenon resulting from cascading effects posed by the regime shift of polythermal glaciers. The Northern Bliggferner Glacier covered the rock slope



before the failure. More than $3.9 \cdot 10^6 \, m^3$ of rock and ice were mobilized by the first time formation of the rock slide in 2007. The rock slope was mechanically impacted by changes in the cryosphere, culminating in the formation of the basal shear zone and several slabs. The creation of a persistent shear zone was favored by the orientation of fault zones and the foliation of the paragneissic rock slope. While multiple processes contributed to the destabilization of the Bliggspitze rock slope, water infiltration due to the increase of the Polythermal Dividing Line/crevasse opening led to the buildup of hydrostatic pressure in the rock mass on a large/zonal limited scale, suggesting the strongest trigger for the first time formation of the rock slide.

The Bliggspitze rock side is not believed to be a singular case. The recent rock slope failure at Piz Scerscen in Switzerland on 14 April 2024 showed characteristics similar to the Bliggspitze rock slide presented here. The affected failure volume was roughly estimated to be about $1 \cdot 10^6 \, m^3$, the west-facing mountain flank is situated in the ice-poor permafrost zone (Kenner et al., 2019) and was previously covered by glacier ice. Before the failure, an anomalous period of warm April weather occurred. The failure scarp exhibited wet patches most likely attributed to zonal infiltration of water. Similar conditions, a warm weather period prior to the failure, and wet areas at the failure scarp have been observed for the Mt. Steller rock-ice avalanche too (Huggel et al., 2008).

A further prominent example of a glacier-induced mass movement is the severe collapse of the Marmolada Glacier (IT) in June 2022. Although the detachment occurred at the glacier-bedrock interfaceand primarily affected the glacier ice, the processes leading to the failure are similar to the combined rock-slope/glacier failures observed Bliggspitze and Piz Scerscen. Marta et al. (2023) hypothesized that the Marmolada glacier collapse was triggered by the buildup of water pressure at the temperate ice/bedrock interface while the cold glacier margins hindered water outflow.

Water may not only act as a trigger for large slope instabilities in cryospheric environments. The abundant amount of water stored not only in liquid form but also in solid form of snow and ice substantially increases run-out length and impacts the mechanics of the propagating rock avalanches as observed in the rock slope failures of Mount Chamoli (IND) in 2021 (Shugar et al., 2021), Fluchthorn (AT) in 2023 (Krautblatter et al., 2024) and Piz Scergen (CH) in 2024. Concomitant with climate warming, we expect further large slope instabilities as higher areas undergo paraglacial transition. Regardless of triggering the failure or increasing its cascading effects, the analysis of water availability is substantial in anticipating large rock slope failure in alpine cryosphere.

## 6 Conclusion

In this study, we developed a conceptual model explaining the uplift of the cold/warm dividing line of polythermal glaciers concomitant to climate warming as a result of the holistic observations of the current and past state of the cryosphere at the field site. A shift of the Polythermal Dividing Line (PDL), a term first defined in this paper, poses thermal and hydrological implications for the rock slope underneath the glacier, thereby challenging its mechanical stability. We simulated the mechanical impact on rock slope during a regime shift of the polythermal Northern Bliggferner Glacier considering glacier mass loss, changing mechanical properties of frozen/unfrozen discontinuities, and adaption of the hydrogeological situation. To



complement the mechanical framework of deep-seated shear planes and zones in permafrost, we tested the tensile strength of anisotropic paragneiss at frozen and unfrozen states. The main findings of the study are presented subsequently.

- Dynamics of glaciers, permafrost, and hydrogeology are strongly interlinked. The Bliggspitze rock slide can be described as a failure type prepared and triggered by these interlinked dynamics. The hydro-mechanical simulations reveal that different processes (glacier unloading, permafrost degradation, rise in groundwater table) affect rock slope stability in different magnitudes. Still, the combination of several simultaneous processes appears to be the most effective method for destabilizing rock slopes, leading to a progressive failure mechanism.

- The most intense impact on destabilizing glacier covered rock slopes in a short time results from the shift of a polythermal glacier towards a temperate regime and its hydrogeological implications. The infiltration of water below areas that were previously sealed with cold-glacier ice substantially alters hydrologeological conditions. As a consequence of snow-, glaciermelt water or rainfall, transient buildup of hydrostatic water pressure in the fracture network may eventually trigger slope failures.

- Glacial unloading affects overall slope stability by the change in the insitu stress field. Stress redistribution leads to crack propagation and can peak in the onset of shearing or even total rock slope failure. Although our model could not prove a distinct decrease in stability by the simulation of glacier unloading due to inherent limitations of the set-up, we found that the rock slope was highly suceptible to glacier erosion, i.e. the removal of rock at the toe of the slope.

- Permafrost thaw destabilizes large rock slides with deep-seated shear zones by mechanical degradation of rock bridges and by its consequences posed on hydrogeology. Ice-filled fractures are less permeable than ice-free fractures and the degradation of ice-infillings additionally releases stored water.

- Tensile strength serves as a proxy for representing strength of rock bridges in a Mohr-Coulomb shear criterion. It was demonstrated that the tensile strength of foliated Paragneis is highly dependent on the frozen/unfrozen state, whereas the foliation orientation has a minor effect on tensile strength.

- Smaller pre-failure events and a warm climate in the months leading up to the failure, which peaked at a historical temperature record, preceded the main failure event at Bliggspitze on 29 June 2007. The timing of the failure coincides with the daily peak river discharge temperature recorded at the Vernagtferner Glacier in close proximity.

- The measured high conductivity of springs below the Bliggspitze rock slide indicates permafrost degradation, as geomorphic analysis has excluded block glaciers as a water source. The loss of cold ice, such as the observed loss of ice aprons in steep faces above the Northern Bliggferner Glacier, has been visually documented from 1969 to 2003. Regarding elevation, aspect, and measured Ground Surface Temperatures, the head scarp resulting from the failure is situated in permafrost-favorable terrain.



*Code availability.*

The authors declare that the code used in this study is available upon reasonable request. Interested researchers can contact the corresponding author to obtain access to the code.

*Data availability.*

    The authors declare that the data used in this study is available upon reasonable request. Interested researchers can contact the corresponding author to obtain access to the data.

*Author contributions.*

    FP wrote the manuscript, conducted the data analysis and drafted the concept for the UDEC modeling study. SW acted as co-pilot and mentor throughout the processes of developing the paper structure and helped to essentially improve the manuscript. JS conducted the tensile strength tests in the laboratory. CZ contributed to frame the setup of the mechanical modeling study and help to revise of the draft of the manuscript. CF recorded the Ground Surface Temperature data used in this study. JF
wrote text regarding the analysis of the Northern Bliggferner Glacier. MK initiated the main concept and first framework for the paper. He conducted the ERT measurement in 2009.

*Competing interests.*

    At least one of the (co-)authors is a member of the editorial board of Earth Surface Dynamics.

*Acknowledgements.* This publication was funded by the Bavarian State Ministry of Science and the Arts through Project IDK M3OCCA. The authors would like to thank Martin Stocker-Waldhuber and Michael Kuhn for providing access to the DHM69 dataset, as well as Christian Sommer for providing the Glacier Thickness Model. In addition, we thank Karl Krainer for providing the publication containing the mapping/measurements of springs.



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
