# Peer review of "Massive permafrost rock slide under a warming polythermal glacier deciphered through mechanical modeling (Bliggspitze, Austria)"

_EGUsphere, 2024_

## Referee Comment (RC1)

Review of the manuscript: egusphere-2024-2509

Massive permafrost rock slide under warming polythermal glacier (Bliggspitze, Austria)

By: Felix Pfluger, Samuel Weber, Joseph Steinhauser, Christian Zangerl, Christine Fey,

Johannes Furst, and Michael Krautblatter

The manuscript is a back analyses of a multimillion cubic meter rock slide at Bliggspitze on 29 June 2007. It is based on a detailed geological/structural model of the failure zone, remote sensing data of the glacial extension of the Bliggferner glacier and its change, on-site temperature data from the failed mountain as well as metrological data from close by meteorological stations, data on springs from the slope, electric resistivity tomography on the slope from after failure as well as laboratory experiments on rock samples from the mountain and advanced 2D stability modelling using the software UDEC.

The topic is extremely timely as it relates slope stability to climate change. This case is particular interesting and unique as it investigates a rockslide that forms subglacial, and the study is unique in its depth in this environment adding to multiple studies in permafrost environment that are less related to glacial ice decay. The manuscript is very well written and balanced. Assumptions are well highlighted, and uncertainties based on the limited amount of data thoroughly discussed. Sensitivity tests were carried out and are well described. This manuscript puts light on changes that will occur in the high alpine but also artic environment with potential hazardous consequences for society.

I can only suggest minor revisions/technical corrections to this well written manuscript with a high quality and well-developed figures.

My suggestions are:

Add "slope stability analysis" into the title.

Add coordinate system in all maps or block diagrams

Add view position of all photos presented into the maps

Add directions to the photos eg. NW-SE in upper right and left

Figure 16 is too small in size in the manuscript, it is difficult to read, consider enlarging

Add more references from outside the Alps to the reference list which will set a more global perspective. E.g. some suggestions:

Line: 33:

Geertsema, M., Menounos, B., Bullard, G., Carrivick, J. L., Clague, J., Dai, C., et al. (2022). The 28 November 2020 landslide, tsunami, and outburst flood–A hazard cascade associated with rapid deglaciation at Elliot Creek, British Columbia, Canada. *Geophysical research letters*, 49(6), e2021GL096716.

Svennevig, K., Hicks, S. P., Forbriger, T., Lecocq, T., Widmer-Schnidrig, R., Mangeney, A., et al. (2024). A rockslide-generated tsunami in a Greenland fjord rang Earth for 9 days. *Science*, 385(6714), 1196-1205.

Kuhn, D., Torizin, J., Fuchs, M., Hermanns, R., Redfield, T., & Balzer, D. (2021). Back analysis of a coastal cliff failure along the Forkastningsfjellet coastline, Svalbard: Implications for controlling and triggering factors. *Geomorphology*, 389, 107850.

Line 39:

Svennevig, K., Dahl-Jensen, T., Keiding, M., Merryman Boncori, J. P., Larsen, T. B., Salehi, S., et al. (2020). Evolution of events before and after the 17 June 2017 rock avalanche at Karrat Fjord, West Greenland – a multidisciplinary approach to detecting and locating unstable rock slopes in a remote Arctic area. *Earth Surf. Dynam.*, 8(4), 1021-1038. doi:10.5194/esurf-8-1021-2020.

Line 41: (Rewrite the paragraph before accordingly)

Ballantyne, C. K., Sandeman, G. F., Stone, J. O., & Wilson, P. (2014). Rock-slope failure following Late Pleistocene deglaciation on tectonically stable mountainous terrain. *Quaternary Science Reviews*, 86, 144-157.

Hermanns, R. L., Schleier, M., Böhme, M., Blikra, L. H., Gosse, J., Ivy-Ochs, S., et al. (2017) 'Rock-Avalanche Activity in W and S Norway Peaks After the Retreat of the Scandinavian Ice Sheet' *Workshop on World Landslide Forum*. Springer, pp. 331-338.

Line 504:

Geertsema, M., Menounos, B., Bullard, G., Carrivick, J. L., Clague, J., Dai, C., et al. (2022). The 28 November 2020 landslide, tsunami, and outburst flood–A hazard cascade associated with rapid deglaciation at Elliot Creek, British Columbia, Canada. *Geophysical research letters*, 49(6), e2021GL096716.

Line 618: (here there are references missing at all) Some suggestions:

Willenberg, H., Evans, K. F., Eberhardt, E., Spillmann, T., & Loew, S. (2008). Internal structure and deformation of an unstable crystalline rock mass above Randa (Switzerland): Part II - Three-dimensional deformation patterns. *Engineering Geology*, 101(1-2), 15-32. doi:http://dx.doi.org/10.1016/j.enggeo.2008.01.016.

Brideau, M.-A., Yan, M., & Stead, D. (2009). The role of tectonic damage and brittle rock fracture in the development of large rock slope failures. *Geomorphology*, 103(1), 30-49. doi:http://dx.doi.org/10.1016/j.geomorph.2008.04.010.

Welkner, D., Eberhardt, E., & Hermanns, R. L. (2010). Hazard investigation of the Portillo Rock Avalanche site, central Andes, Chile, using an integrated field mapping and numerical modelling approach. *Engineering Geology*, 114(3-4), 278-297. doi:http://dx.doi.org/10.1016/j.enggeo.2010.05.007.

Brideau, M.-A., & Stead, D. (2012). Evaluating kinematic controls on planar translational slope failure mechanisms using three-dimensional distinct element modelling. *Geotechnical and Geological Engineering*, 30, 991-1011.

Lines 775-785:

I would also suggest discussing against:

Geertsema, M., Menounos, B., Bullard, G., Carrivick, J. L., Clague, J., Dai, C., et al. (2022). The 28 November 2020 landslide, tsunami, and outburst flood–A hazard cascade associated with rapid deglaciation at Elliot Creek, British Columbia, Canada. *Geophysical research letters*, 49(6), e2021GL096716.

Svennevig, K., Hicks, S. P., Forbriger, T., Lecocq, T., Widmer-Schnidrig, R., Mangeney, A., et al. (2024). A rockslide-generated tsunami in a Greenland fjord rang Earth for 9 days. *Science*, 385(6714), 1196-1205.

Some minor typos:

Line 307: there seems to be one or more words missing

Line 317: a space missing after "glacier,"

Line 319: consider "up-glacier" and "fracture band"

Line 330: is rather a repetition of the method section

Line 386: May be rewrite "The picture …"

Line 412: Figure number is missing.

Line 451 and 454: consider writing in the same wording: "ice-free stage" and "glacier-free conditions" reads as if different aspects are meant. If indeed different aspects are meant make clearer the difference in both sentences.

Line 459 a space is missing after the bracket

Line 804 the "," should be positioned prior to the line break

---

## Author Comment (AC1)

Answer to https://doi.org/10.5194/egusphere-2024-2509-RC1

Dear Reginald Hermanns,

Thank you for taking the time to review our article and the constructive feedback and input on the previous submission. We addressed all the issues raised and believe that the implemented changes have substantially improved the revised manuscript. We briefly outline the primary changes here and add a one-by-one reply to the reviews on the following pages.

- We updated the title to 'Mechanical modeling deciphering the massive permafrost rock slide under a warming polythermal glacier (Bliggspitze, Austria),' emphasizing the main methods used in the publication.
- Figure sizes and coordinates to maps were adjusted accordingly.
- Thank you for providing publications beyond ours to enhance a global perspective. We included various suggestions and added additional references, which extended and essentially improved the section introduction.

Please find all responses to each comment below, marked in blue.

With best regards,
Felix Pfluger
On behalf of all authors
* * *
Review of the manuscript: egusphere-2024-2509

Massive permafrost rock slide under warming polythermal glacier (Bliggspitze, Austria)

By: Felix Pfluger, Samuel Weber, Joseph Steinhauser, Christian Zangerl, Christine Fey,

Johannes Furst, and Michael Krautblatter

The manuscript is a back analyses of a multimillion cubic meter rock slide at Bliggspitze on 29 June 2007. It is based on a detailed geological/structural model of the failure zone, remote sensing data of the glacial extension of the Bliggferner glacier and its change, on-site temperature data from the failed mountain as well as metrological data from close by meteorological stations, data on springs from the slope, electric resistivity tomography on the slope from after failure as well as laboratory experiments on rock samples from the mountain and advanced 2D stability modelling using the software UDEC.

The topic is extremely timely as it relates slope stability to climate change. This case is particular interesting and unique as it investigates a rockslide that forms subglacial, and the study is unique in its depth in this environment adding to multiple studies in permafrost environment that are less related to glacial ice decay. The manuscript is very well written and balanced. Assumptions are well highlighted, and uncertainties based on the limited amount of data thoroughly discussed. Sensitivity tests were carried out and are well described. This manuscript puts light on changes that will occur in the high alpine but also artic environment with potential hazardous consequences for society.

I can only suggest minor revisions/technical corrections to this well written manuscript with a high quality and well-developed figures.

My suggestions are:

Add "slope stability analysis" into the title.
We updated the title to 'Massive permafrost rock slide under a warming polythermal glacier deciphered through mechanical modeling (Bliggspitze, Austria),' emphasizing the main methods used in the publication -> mechanical modeling.

Add coordinate system in all maps or block diagrams
We described the exact location in the text Bliggspitze summit (3453 m asl, 46°55'5''N, 10°47'10''E - WGS84). We added CRS to Fig. 1 and 16.

Add view position of all photos presented into the maps. Included in Fig. 1.

Add directions to the photos eg. NW-SE in upper right and left Included in Fig. 1.

Figure 16 is too small in size in the manuscript, it is difficult to read, consider enlarging
Corrected.

Add more references from outside the Alps to the reference list which will set a more global perspective. E.g. some suggestions:
The blue-marked references are now included in the manuscript.

Line: 33:

Geertsema, M., Menounos, B., Bullard, G., Carrivick, J. L., Clague, J., Dai, C., et al. (2022). The 28 November 2020 landslide, tsunami, and outburst flood–A hazard cascade associated with rapid deglaciation at Elliot Creek, British Columbia, Canada. *Geophysical research letters*, 49(6), e2021GL096716.

Svennevig, K., Hicks, S. P., Forbriger, T., Lecocq, T., Widmer-Schnidrig, R., Mangeney, A., et al. (2024). A rockslide-generated tsunami in a Greenland fjord rang Earth for 9 days. *Science*, 385(6714), 1196-1205.

Kuhn, D., Torizin, J., Fuchs, M., Hermanns, R., Redfield, T., & Balzer, D. (2021). Back analysis of a coastal cliff failure along the Forkastningsfjellet coastline, Svalbard: Implications for controlling and triggering factors. *Geomorphology*, 389, 107850.

Line 39:

Svennevig, K., Dahl-Jensen, T., Keiding, M., Merryman Boncori, J. P., Larsen, T. B., Salehi, S., et al. (2020). Evolution of events before and after the 17 June 2017 rock avalanche at Karrat Fjord, West Greenland – a multidisciplinary approach to detecting and locating unstable rock slopes in a remote Arctic area. *Earth Surf. Dynam.*, 8(4), 1021-1038. doi:10.5194/esurf-8-1021-2020.
Included: 'Due to the complexity of large rock slope failures, their often remote alpine locations with challenging accessibility that result in limited available data, and the absence of a clear seasonal pattern, predicting such failures in permafrost regions is challenging \citep{Huggel.2008, Svennevig.2020}.'

Line 41: (Rewrite the paragraph before accordingly)

Ballantyne, C. K., Sandeman, G. F., Stone, J. O., & Wilson, P. (2014). Rock-slope failure following Late Pleistocene deglaciation on tectonically stable mountainous terrain. *Quaternary Science Reviews*, 86, 144-157.

Hermanns, R. L., Schleier, M., Böhme, M., Blikra, L. H., Gosse, J., Ivy-Ochs, S., et al. (2017) 'Rock-Avalanche Activity in W and S Norway Peaks After the Retreat of the Scandinavian Ice Sheet' *Workshop on World Landslide Forum*. Springer, pp. 331-338.

We added: Investigating the timing of rock slope failures in Scotland and northwest Ireland in relation to deglaciation, \cite{Ballantyne.2014} found that 95\% of the analyzed failures occurred within approximately 5400 years after deglaciation, with peak activity occurring between 1600 and 1700 years post-deglaciation. They suggests that glacier unloading and seismic activity were the primary triggers. Similarly, \cite{Hermanns.2017} identified a time cluster of rock avalanche deposits in Norway originating from the first millennium after deglaciation, as well as a second cluster during the Holocene climatic optimum. Studying rock slope failures in the European Alps, \cite{Prager.2008} noted a time cluster of events several thousand years after ice withdrawal. These failures, occurring several thousand years after deglaciation, are hypothesized to have been prepared by glacial cycles and finally triggered with a time lag relative to the Last Glacial Maximum (LGM), accounting for the loss of permafrost \citep{McColl.2012, Krautblatter.2013}.

Line 504:

Geertsema, M., Menounos, B., Bullard, G., Carrivick, J. L., Clague, J., Dai, C., et al. (2022). The 28 November 2020 landslide, tsunami, and outburst flood–A hazard cascade associated with rapid deglaciation at Elliot Creek, British Columbia, Canada. *Geophysical research letters*, 49(6), e2021GL096716.

Line 618: (here there are references missing at all) Some suggestions:

Willenberg, H., Evans, K. F., Eberhardt, E., Spillmann, T., & Loew, S. (2008). Internal structure and deformation of an unstable crystalline rock mass above Randa (Switzerland): Part II - Three-dimensional deformation patterns. *Engineering Geology*, 101(1-2), 15-32. doi:http://dx.doi.org/10.1016/j.enggeo.2008.01.016.

We assumed it was state-of-the-art in rock slope mechanics and there is no need to references. Due to the broad readership it does definitely make sense! And I adopted the references and added Eberhardt et al. (2004)

Brideau, M.-A., Yan, M., & Stead, D. (2009). The role of tectonic damage and brittle rock fracture in the development of large rock slope failures. *Geomorphology*, 103(1), 30-49. doi:http://dx.doi.org/10.1016/j.geomorph.2008.04.010.

Welkner, D., Eberhardt, E., & Hermanns, R. L. (2010). Hazard investigation of the Portillo Rock Avalanche site, central Andes, Chile, using an integrated field mapping and numerical modeling approach. *Engineering Geology*, 114(3-4), 278-297. doi:http://dx.doi.org/10.1016/j.enggeo.2010.05.007.

Added Eberhardt et al. (2004):
Eberhardt, E., Stead, D., and Coggan, J.: Numerical analysis of initiation and progressive failure in natural rock slopes—the 1991 Randa885
rockslide, International Journal of Rock Mechanics and Mining Sciences, 41, 69–87, 2004.

Brideau, M.-A., & Stead, D. (2012). Evaluating kinematic controls on planar translational slope failure mechanisms using three-dimensional distinct element modelling. *Geotechnical and Geological Engineering*, 30, 991-1011.

Lines 775-785:

I would also suggest discussing against:

Geertsema, M., Menounos, B., Bullard, G., Carrivick, J. L., Clague, J., Dai, C., et al. (2022). The 28 November 2020 landslide, tsunami, and outburst flood–A hazard cascade associated with rapid deglaciation at Elliot Creek, British Columbia, Canada. *Geophysical research letters*, 49(6), e2021GL096716.
This case is an example of a valley glacier in contact with the toe of the rock slope. The glacier for the case of Elliot Creek was only covering marginal parts of the lower part of the rock slope. Its thermal impact on the rock slope is not of central interest as it is the case for Marmolata glacier (Chiarle et al. 2023), Mt. Steller (Huggel et al. 2008), or here at Bliggspitze.

Svennevig, K., Hicks, S. P., Forbriger, T., Lecocq, T., Widmer-Schnidrig, R., Mangeney, A., et al. (2024). A rockslide-generated tsunami in a Greenland fjord rang Earth for 9 days. *Science*, 385(6714), 1196-1205.

-> See comment before.

Some minor typos:
Thank you, we corrected all the typos!

Line 307: there seems to be one or more words missing

Line 317: a space missing after "glacier,"

Line 319: consider "up-glacier" and "fracture band"

Line 330: is rather a repetition of the method section

Line 386: May be rewrite "The picture ..."

Line 412: Figure number is missing.

Line 451 and 454: consider writing in the same wording: "ice-free stage" and "glacier-free conditions" reads as if different aspects are meant. If indeed different aspects are meant make clearer the difference in both sentences.

Line 459 a space is missing after the bracket

Line 804 the "," should be positioned prior to the line break

---

## Author Comment (AC2)

Answer to https://doi.org/10.5194/egusphere-2024-2509-CC1
and https://doi.org/10.5194/egusphere-2024-2509-RC2

Dear Philip Deline,

Thank you for taking the time to review our article and the constructive feedback and input on the previous submission. We addressed all the issues raised and believe that the implemented changes have substantially improved the revised manuscript. We briefly outline the primary changes here and add a one-by-one reply to the reviews on the following pages.

- We updated the description of the failure chronology (pre-, sudden-,post-failure) and its placement in the manuscript.
- We added the correct volumina to the Piz Scescen rock slide, as stated by the Permos Rock Fall Data Portal.
- We adopted your suggestions for rephrasing the sub-sections to enhance its connection to the content described.
- We apologize for the numerous typos you highlighted; we have corrected them all.

Please find all responses to each comment below, marked in blue.

With best regards,
Felix Pfluger
On behalf of all authors
* * *
**Comments by P. Deline to Pfluger et al.**

"Massive permafrost rock slide under warming polythermal glacier (Bliggspitze, Austria)"

Submitted to *Earth Surface Dynamics*

Pfluger et al. proposed in "Massive permafrost rock slide under warming polythermal glacier (Bliggspitze, Austria) " to explore how far change in the thermal regime at the base of a mountain glacier, from cold- to warm-based, in relation with the degradation of the permafrost that affects/affected the glacier bedrock, can be a triggering factor of rock slope failure. They built their study on a large rock slide that occurred on a glaciated slope in Austria in June 2007. They analysed glacier changes since the 1970s, ground surface and in depth temperatures, and water flow source by combining analysis of pre- and post-failure various datasets (orthoimages and DEMs, meteorological and hydrological data, mineralization of springs, GST), performing ERT survey and rock testing in lab, and modelling the mechanical impact of hydrostatic pressure, degradation of permafrost and glacier retreat on rock slope stability.

Pfluger et al. developed four individual scenarios focusing on: 1) structural predisposition, 2) glacier unloading, 3) permafrost degradation, and 4) peak groundwater level. These conceptual scenarios allow them to investigate rock slope mechanics and interdependencies with environmental forcings in a UDEC model.

They conclude that glacier unloading, permafrost degradation and rise in groundwater table, related to the uplift of the cold/warm dividing line in the polythermal glacier, are strongly interlinked. Glacial unloading affects slope stability, permafrost thaw degrades rock bridges (whose tensile strength) and ice-filled fractures, while the resulting hydrostatic water pressure in the fracture network likely triggered the slope failure. However, as stated on L614-615:

"hydrogeology in fractured permafrost rock mass is scarcely researched and explained mainly by conceptual considerations rather than by real in-situ observations".

The manuscript represents a relevant contribution to scientific progress within the scope of the journal, exploring the impact of uplift of the cold/warm dividing line of polythermal alpine glaciers on rock slope stability. Substantial conclusions are reached.

The results supporting the interpretation and conclusions are discussed in an appropriate and balanced way, the assumptions are clearly outlined, while the limitations of the modelling are underlined. The authors clearly indicate their own original contribution.

The manuscript is presented in a clear and well-structured way, with many nice and expressive figures; however, symbols and abbreviations in some figures should be explained in their captions. The use of English language is appropriate as far as I can tell. Below my general comment is a long list of typos and small carelessnesses that a careful proofreading would have allowed for correction.

While the abstract provides a complete summary, I would suggest a slight change in the title: "Massive rock slide in permafrost-affected slope under warming polythermal glacier (Bliggspitze, Austria)". References are appropriated, as is the supplementary material.

After further discussion, we have opted to retain the phrasing "permafrost rock slide" in the title. We believe it maintains a balance between precision and readability for our intended audience.

The following list points out, line by line, on one hand typos and small negligences, and on the other hand some remarks and suggestions:

L36-39: this paragraph starts about "zones of potential permafrost" but ends with "elevations below 2000 m asl" where permafrost (at the core of the concluding sentence that follows) is absent. For clarity, please reorganize this paragraph.

For clarity, we now refer only to failures in the zone of permafrost and dropped the additional information, as it might be misleading and is of minor importance.

L47: "stress relief": stress **release**? Corrected.

L58: add "according to" before Mamot et al. (2018). Corrected.

L71: correct "degra**d**ation". Corrected.

L74: replace (AT) by (Austria). Corrected.

L104: "Post-failure activity was evident in the months and years after the initial rock slope deformation". There is a possible confusion for the reader between "initial rock slope deformation", and "first time formation of the rock slide". I had not clearly understood the difference between both, respectively your at-failure and pre-failure stages, before reading L126-128. So I would suggest to introduce the first paragraph of section 3.1 earlier in the text, as these terms are used before the section 3.1.

For clarity, we placed the definition of failure stages into the introduction and rephrased the terms in this chapter.
'To distinguish between the phases of slope deformation, we define the pre-failure, sudden-failure, and post-failure stages as follows: The pre-failure stage spans months, years, or even decades, during which the rock slope undergoes initial deformation, typically at a rate of

millimeters to centimeters per year, before experiencing a distinct acceleration. The sudden-failure stage marks the first time formation of the Bliggspitze rock slide in June 2007, coinciding with the creation of the basal shear zone at the head scarp. This period is characterized by a sharp increase in velocity, reaching from meters up to several decameters per day, and associated activity in the hours and days surrounding the event. The post-failure stage involves the ongoing deformation of the displaced rock mass in the months and years following the first time formation of the rock slide \citep{Leroueil.1996}.'

Note that on L149 and 346, you wrote: "on 12 August 2009, two years after **the first time formation of the Bliggspitze rock slide**", that is a confusing as "the first time formation" doesn't corresponds to the June 2007 rock slide but is predating this event (pre-failure stage)…
First time formation of rock slide is the sudden-failure stage which is now correct after the updated definition.

Same confusion on L763: "More than 3.9 · $10^6 m^3$ of rock and ice were mobilized by **the first time formation of the rock slide** in 2007"…
See previous comment.

May be you should invert the expressions, as : initial rock slope deformation = pre-failure stage, and first time formation of the rock slide = at-failure stage, i.e. days/hours before and after the June 2007 rock slide. To me, it sounds better…
We adopted this approach and included it in the definition, adding also velocities of mass movement for more precise definitions.

L110: give the meaning of the acronym ALS (i.e. airborne laser scanning). Corrected.

Fig.1: I suggest to change the black-dashed line of the rectangle on the left map corresponding to the zoom on the right map, as this black-dashed line can be confused with the one of the glacier transect. Corrected.

In the legend of the left map (and on L161), correct "t**h**reshold". Corrected.

L125: "3.1 Pre-failure analysis of the Bliggspitze rock slide". This section is describing also post-failure data, in order to realise a pre-failure analysis, so may be complete its title as: "3.1 **Field methods and data for** pre-failure analysis of the Bliggspitze rock slide"? By the way, "Pre-failure analysis of the Bliggspitze rock slide" is the (correct) title of section 4.1.
True. It covers more than that.

Table 1: give the meaning of the acronyms T, PPT, Q. Corrected.

L134 and 136: keep 'stage', used earlier, rather than 'stadium'. Corrected.

L145: add "and" between "Haberkorn et al. (2015) **and** Draebing et al. (2022)". Corrected.

L157-158: "Northern and Southern Bliggferner and Eiskastenferner Glaciers". Before this occurrence (and on Fig.3), the glacier names were without a G, and with G after. Please normalize everywhere in the text. Corrected.

L162-163: I suggest you move the sentence "The chronology preceding the formation of the Bliggspitze rock slide was analyzed by utilizing climate, meteorological, and discharge data and examining the related events that occurred in the years and weeks before the rock slide (Table 1)." to the beginning of the section 3.1. Corrected.

L207: 3200, not 3**.**200 m Corrected.

L244: (**Fig.** 2a), not (2a). Corrected.

Fig. 3: give the meaning of all the acronyms and symbols in the caption. Corrected.

L296: cryosphere, not cryo**ps**here. Corrected.

L307: complete with "post-failure period" between "the" and "is". Corrected.

L311: complete (Fig. 4**b**) Corrected.

L315: delete a in (Fig. 5a). Corrected.

L317: replace "are" by "is". Corrected.

L318: correct "transverese". Corrected.

Fig. 4: in the caption, replace S4 by **S3** in the sentence "The trace of cross-section is displayed in Figure 1 and S4". Corrected.

L339: add °C to "0.5**°C**". Corrected.

Fig.7: in one of the 'green' boxes, correct "relativ**ely** dry". Corrected.

L355-356: the sentence "The interpretation of subsurface areas is based on resistivity values suggested by Krautblatter and Hauck (2007); Keuschnig et al. (2017); Offer et al. (2024)" could be deleted as it repeats the one in the caption of Fig.7. Corrected.

Fig. 8: in its legend box, the light green bar for "slope,proximal" has disappeared. Corrected.

L368: correct "a 227-year climate record" by "a 22**8**-year climate record", as it is in the caption of Fig.9. Corrected.

Fig.9: give in the caption the meaning of the symbols used in the graph. Corrected.

L374: replace "They confirmed…" by "**We** confirmed…"? Corrected.

L393: "paragneis**sic**" Corrected.

Fig.11: a) and b) are missing on the figure. Corrected.

L412 and in caption of Fig.15: complete "(see also Fig. ??) and "Figure ??" Corrected.

L422-423: delete "shown in the supplementary material." Corrected.

Fig.13: in its caption, correct "montiroing" by "monitoring". Corrected.

L469-470: correct "(Fig. 13b and c)" with "(Fig. **12**b and c)". Corrected.

Fig.15: correct "verus" in the 1ˢᵗ line of the caption, and complete "Figure **??**" Corrected.

L505: add "according to" before "Faillettaz et al. (2015)". Corrected.

L510: delete "could" or replace "could show" by "suggest"? Corrected.

L519: add "according to" before "Irvine-Fynn et al. (2011)". Corrected.

L522: delete one of the "the". Corrected.

L528-529: reorganize the sentence "Ground Surface Temperatures generally reveal permafrost favourable conditions at the elevation of the head scarp conducted in the years after the failure" as follows: "Ground Surface Temperatures **conducted in the years after the failure** generally reveal permafrost favourable conditions at the elevation of the head scarp". Corrected.

Fig.16: in its caption, correct "di**s**cussion" and "abbreviation**s**". Corrected.

Besides, I suggest to explain the symbols and abbreviations in the caption, and to enlarge a bit the figure. We included a short "caption-storyline," including an explanation of symbols to enhance the understanding of the

L550: add "according to" before "Noetzli and Gruber (2009)" Corrected.

L554: "the warming of air temperatures": as T neither heats nor cools, replace by "the warming of air". Corrected.

L558: correct "clef**t**-ice". Corrected.

L578-579: add comma before "(ii)" and "or (iii)". Corrected.

L582: replace (GER) by (Germany). Corrected.

L583: correct "resutled" with "resu**lt**ed". Corrected.

Add "according to" before "Scandroglio et al. (2024)." Corrected.

L586-587: add "according to" before "Caduff et al. (2021); Hilger et al. (2021); Etzelmuller et al. (2022)", or put in parentheses. Corrected.

(I WILL NOT MENTION THIS RECURRING DEFECT IN THE REST OF THE TEXT… PLEASE CHECK IT) Corrected.

L633: correct "in**id**cate". Corrected.

L634: correct "wheatered" Corrected.

L640: add "in Table 3" in "(see scenario S1 geometry C **in Table 3**). Corrected.

L646: correct "Galcial" Corrected.

L649: replace ";" by "and". Corrected.

L651: replace 5 by **14** in "(Fig. **5**: stage 9)" Corrected.

L669: the sub-section title "**S3 - Permafrost distribution**" doesn't cover all the elements discussed below. Please complete it.
the discussion is on permafrost affecting deep-seated shear planes. We adapted the sub-section title: S3 - Permafrost distribution and its effect on deep-seated shear planes

L669-670: "By iteratively incrementing the mountain permafrost altitude, the resulting rock slope displacement increased from a minimum of 0.1 m to a maximum of 0.08 m." : if 0.1 m is a minimum, 0.08 m can't be a maximum, or your sentence has to be rephrased (values related to the different MPA?). Typo, now corrected and reference to Figure 15 included.

Besides, from where these two values are coming, as they are not in the Results section? Reference to Figure 15 included.

L713: correct "existance". Corrected.

L739-740: rephrase a bit the sentence (by deleting **impacts**?): "The results of the BZT do not indicate that the foliation of intact rock impacts is structurally controlling the rock slope mechanics." Corrected.

L753: delete the final '**s**' on "result**s**". Corrected.

L768-770: "Piz Scerscen [...] The affected failure volume was roughly estimated to be about 1 · $10^6$ m³". May be this estimate could be revised, as I was told that this V could reach 5 Mio m³. And more than wet patches in the failure scarp, water flow was observed in the following hours. S. Weber and other colleagues at SLF Davos for instance could update the elements about 2024 Piz Scersen rock-ice avalanche.
We updated the correct volume that was about 5 mio. m³ (https://www.permos.ch/data-portal/rock-falls). More detailed information on water flow is missing according to our knowledge.

L775: rather than "a glacier-induced mass movement is the severe collapse of the Marmolada Glacier (IT)", should not be more correct to call it a water pressure-induced mass movement? I agree, it is the glacier mass the moves, but movement was induced/triggered by the water pressure.

Correct **IT** with **Italy**. Corrected.

L778: correct « Marta et al. (2023)" with **Chiarle** et al., as Marta is the first name of Chiarle. And correct it too on L1000 in the References. Corrected.

L782: write IND, AT and CH in full. Corrected.

L785: complete as follows "the analysis of water, **ice and snow** availability" Corrected.

L807: correct "altough". Corrected.

L820: "block glaciers"? Are you meaning **rock** glaciers (among the other excluded sources)? Yes, rock glacier; I confused it with the german word 'Blockgletscher'.